# Quantifying the causal impact of biological risk factors on healthcare costs

Jiwoo Lee [1,2,3,4], Sakari Jukarainen[2], Antti Karvanen[2], Padraig Dixon[5], Neil M. Davies [6,7,8,9], George Davey Smith [9], Pradeep Natarajan [1,3,4,10] & Andrea Ganna [1,2,11] ✉

Understanding the causal impact that clinical risk factors have on healthcare-related costs is critical to evaluate healthcare interventions. Here, we used a genetically-informed design, Mendelian Randomization (MR), to infer the causal impact of 15 risk factors on annual total healthcare costs. We calculated healthcare costs for 373,160 participants from the FinnGen Study and replicated our results in 323,774 individuals from the United Kingdom and Netherlands. Robust causal effects were observed for waist circumference (WC), adult body mass index, and systolic blood pressure, in which a standard deviation increase corresponded to 22.78% [95% CI: 18.75-26.95], 13.64% [10.26-17.12], and 13.08% [8.84-17.48] increased healthcare costs, respectively. A lack of causal effects was observed for certain clinically relevant biomarkers, such as albumin, C-reactive protein, and vitamin D. Our results indicated that increased WC is a major contributor to annual total healthcare costs and more attention may be given to WC screening, surveillance, and mitigation.

Healthcare costs continue to rise worldwide, and in 2018, global healthcare spending reached $8.3 trillion, or 10% of the global gross domestic product[1]. While healthcare costs continue to rise, morbidity is rising, so better understanding of healthcare costs and cost efficiency is critical[1]. Accurate measurement of healthcare costs caused by different risk factors and health outcomes is important to prioritize public health promotion and prevention programs[2]. Moreover, healthcare costs can act as a proxy of disease burden when investigating the effects of risk factors. Thus, epidemiology, public health, and policy stakeholders are very interested in the analysis of healthcare costs[3].

Several studies have quantified the healthcare costs associated with different risk factors[4,5]. For example, Bolnick et al.[4] calculated the correlation between United States healthcare spending and 84 modifiable risk factors from the Global Burden of Disease study, and

Goetzel et al.[5] calculated the correlation between healthcare costs and 10 modifiable risk factors including blood glucose, obesity, stress, depression, and physical inactivity. However, there are several limitations with such studies. First, associations between risk factors and healthcare burden are based on observational data and suffer from challenges such as confounding and reverse causation. Second, most studies do not estimate the direct association between risk factors and healthcare costs, but first estimate the impact of risk factors on different diseases and subsequently link each disease to estimated healthcare costs[4,6,7]. Thus, the impact of risk factors on healthcare costs that are not directly captured by diseases (e.g., medications) were not considered. Third, while modifiable risk factors such as smoking and alcohol consumption have been studied[8], little is known about the impact on healthcare costs of commonly measured

[1]Broad Institute of MIT and Harvard, Cambridge, MA, USA. [2]Institute for Molecular Medicine Finland (FIMM), HiLIFE, University of Helsinki, Helsinki, Finland. [3]Program in Medical and Population Genetics and the Cardiovascular Disease Initiative, Broad Institute of Harvard and MIT, Cambridge, MA, USA. [4]Cardiovascular Research Center and Center for Genomic Medicine, Massachusetts General Hospital, Boston, MA, USA. [5]Nuffield Department of Primary Care Health Sciences, University of Oxford, Oxford, UK. [6]Division of Psychiatry, University College London, Maple House, 149 Tottenham Court Rd, London W1T 7NF, UK. [7]Department of Statistical Sciences, University College London, London WC1E 6BT, UK. [8]K.G. Jebsen Center for Genetic Epidemiology, Department of Public Health and Nursing, Norwegian University of Science and Technology, Trondheim, Norway. [9]Medical Research Council Integrative Epidemiology Unit, University of Bristol, BS8 2BN Bristol, UK. [10]Department of Medicine, Harvard Medical School, Boston, MA, USA. [11]Analytic and Translational Genetics Unit, Massachusetts General Hospital, Boston, MA, USA. ✉e-mail: andrea.ganna@helsinki.fi

biomarkers, which are generally the direct targets of pharmacological interventions.

An alternative source of evidence to assess the effects of diseases and biomarkers on healthcare costs is Mendelian Randomization (MR), which addresses some of the previous limitations. MR is a method that uses genetic variants as instrumental variables to estimate causal relationships between exposures and outcomes and can address the issues of confounding and reverse causation[9]. MR is particularly powerful for estimating the effects of biological risk factors with a strong genetic bases, such as clinical biomarkers and biometrics, including body mass index and blood pressure.

Previous studies have used MR to identify the causal effects of adiposity[10], body mass index[11,12], and common health conditions[13]. However, these studies were either based in the UK Biobank (e.g., limited to relatively healthy individuals between 40 and 69 years old) or did not have complete coverage of healthcare costs associated with medication and primary care costs. No studies to date have used MR to comprehensively link a diverse set of biological risk factors to healthcare costs. In this study, we used a large prospective study from Finland, the FinnGen Study, with genetic information available for 373,160 individuals linked to several national healthcare registries covering primary, secondary, and medication costs. Because of the high-quality, long follow-up, and detailed healthcare costs available in these registries, we were able to obtain an accurate and comprehensive estimate of annual healthcare expenditure. We further assessed the generalizability and robustness of our findings by accounting for selection bias and by leveraging additional healthcare cost data from 323,774 individuals from the United Kingdom and Netherlands.

In this study, we (1) evaluated the causal impact of 15 risk factors, with strong genetic bases, on annual total healthcare costs, (2) identified whether the effects vary by service type, age, and sex, and (3) quantified the mediating of effects of major diseases. We show that elevated waist circumference, adult body mass index, and systolic blood pressure are major causal contributors to healthcare costs within a causal inference framework.

## Results

In this study (Fig. 1), we estimated the causal impact of 15 risk factors with strong genetic bases (Supplementary Table 1) on annual total healthcare costs.

## Distribution of healthcare costs

We included 373,160 FinnGen participants (data freeze 8) followed-up to a maximum of 22 years. The average age at baseline (i.e., date of DNA sample collection) was 54 years old and 56% of the study cohort was female. The mean and median annual total healthcare cost was €2,706 and €1,313, respectively (Fig. 2). Primary care (mean = €169, median = €109) and medication (mean = €518, median = €202) costs were lower than secondary care (mean = €2019, median = €852) costs. Mean (females = €2244, males = €3303) and median (females = €1245, males = €1433) costs were similar in male and females but males (SD = €15545) had greater variability than females (SD = €4445). Individuals over the age of 60 (mean = €3406, median = €1800) had greater healthcare costs than individuals between the age of 30 and 60 (mean = €1851, median = €891) and individuals under the age of 30 (mean = €1484, median = €621). GWAS performed on log-transformed annual total healthcare costs identified several genome-wide significant SNPs (Supplementary Table 2 and Supplementary Figs. 1, 2).

## Causal impact of risk factors on total healthcare costs

We estimated the causal impact of risk factors on healthcare costs using MR. All risk factors had strong genetic instruments (e.g., F-statistic > 50) obtained from genome-wide association studies of at least 173,082 individuals. We detected significant effects of six risk factors on costs (i.e., waist circumference, adult body mass index, systolic blood pressure, triglycerides, cystatin C, and HDL cholesterol) at the Bonferroni-corrected significance level ($P < 3.33 \times 10^{-3}$) (Fig. 3). We performed sensitivity analyses using five different robust MR approaches (Supplementary Table 3) and identified three risk factors that consistently affected total annual healthcare costs across at least three of the sensitivity analyses: waist circumference (WC), adult body mass index (BMI), and systolic blood pressure (SBP). One standard deviation (SD) increase in WC increased the annual total healthcare costs by 22.78% (95% CI: [18.75, 26.95], $P = 1.90 \times 10^{-33}$); one SD increase in adult BMI increased the annual total healthcare costs by 13.64% (95% CI: [10.26, 17.12], $P = 1.06 \times 10^{-16}$); and one SD increase in SBP increased the annual total healthcare costs by 13.08% (95% CI: [8.84, 17.48], $P = 2.80 \times 10^{-10}$). Using MR methods robust to pleiotropy, we found similar effects for WC (17.19–22.78%), adult BMI (8.21–13.64%), and SBP (13.08%–32.36) (Supplementary Table 3).

Fig. 1 | Graphical abstract. a Example of how genetic variants associated with BMI and randomly assigned at birth can be used to infer the causal impact of BMI on healthcare costs (e.g., by modifying risk for cardiovascular disease and statin medication). b Assumptions underlying MR. 1: Genetic instruments must be robustly associated with the exposure (risk factor), 2: there must be no confounders of the genetic instruments-outcome association, and 3: Genetic instruments must not influence the outcome except through the exposure. c National healthcare registries link with the FinnGen Study to estimate annual total healthcare costs. d STROBE flow diagram for study cohort, in which 373,160 individuals were included.

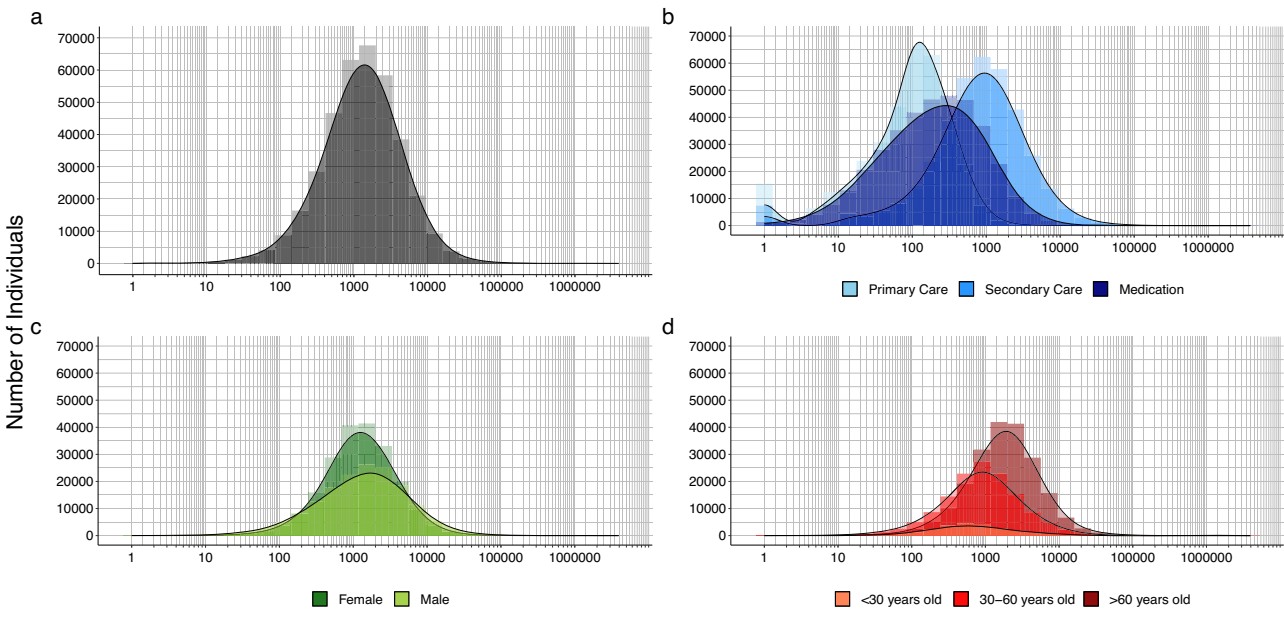

**Fig. 2 | Distribution of healthcare costs in 373,160 FinnGen participants.**
**a** Annual total healthcare cost in euros. **b** Annual healthcare costs in euros for primary care, secondary care, and medication costs. **c** Annual total healthcare costs in euros for females and males. **d** Annual total healthcare costs for individuals under 30 years old, between 30 and 60 years old, and over 60 years old. X-axis is on a log10-transformed scale.

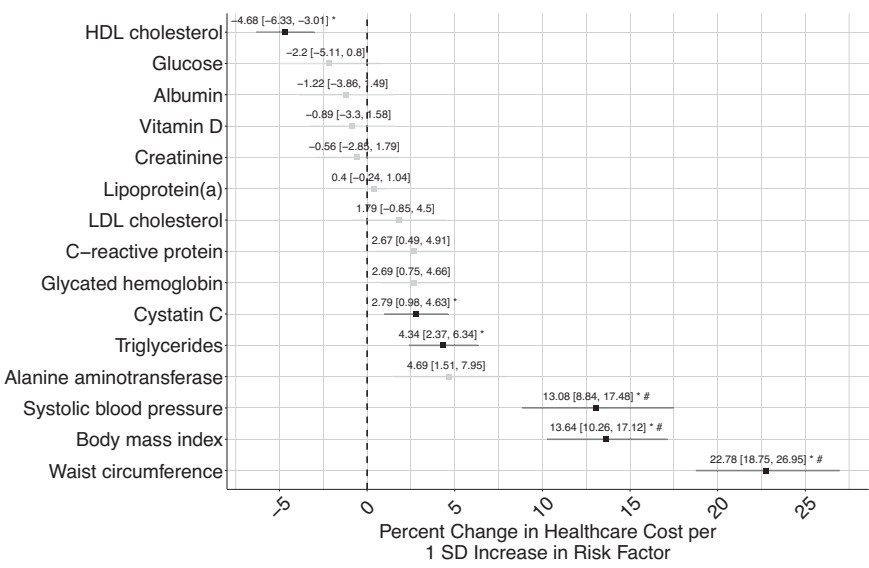

**Fig. 3 | Mendelian randomization on 15 biological risk factors on annual total healthcare costs for 373,160 FinnGen participants using the two-sample, inverse variance weighted approach.** HDL cholesterol ($P = 7.23 \times 10^{-8}$), cystatin C ($P = 2.35 \times 10^{-3}$), triglycerides ($P = 1.24 \times 10^{-5}$), systolic blood pressure ($P = 2.80 \times 10^{-10}$), body mass index ($P = 1.06 \times 10^{-16}$), and waist circumference ($P = 1.90 \times 10^{-33}$) had a significant, causal effect on annual total healthcare costs. Two-sided $p$ values were calculated from the effect estimates and standard errors of the Mendelian Randomization model and adjusted for multiple hypothesis testing. Bars indicate 95% confidence interval. Black bars and the * symbol indicate biological risk factors that are statistically significant at the Bonferroni-corrected significance level ($P < 3.33 \times 10^{-3}$). The # symbol indicates biological risk factors that were significant across at least three of the MR approaches used in sensitivity analyses. SD is standard deviation. Source data are provided as a Source Data file.

Several biomarkers did not have a significant (e.g., Bonferroni-corrected significance level of $P < 3.33 \times 10^{-3}$) impact on annual total healthcare costs (e.g., alanine aminotransferase, $P = 4.58 \times 10^{-2}$; glycated hemoglobin, $P = 6.44 \times 10^{-3}$; C-reactive protein, $P = 1.64 \times 10^{-2}$; LDL cholesterol, $P = 1.86 \times 10^{-1}$; lipoprotein(a), $P = 2.20 \times 10^{-1}$; creatinine, $P = 6.39 \times 10^{-1}$; vitamin D, $P = 4.75 \times 10^{-1}$; albumin, $P = 3.73 \times 10^{-1}$; glucose, $P = 1.50 \times 10^{-1}$), indicating that genetically-increased levels of these biomarkers do not result in a significant downstream impact on healthcare costs. LDL cholesterol (1.79%, 95% CI: [−0.85, 4.50], $P = 1.86 \times 10^{-1}$) had a null effect on healthcare costs, despite the strong genetic instruments for LDL cholesterol. We performed sensitivity analyses using genetic instruments for triglycerides, HDL cholesterol, and LDL cholesterol that were adjusted for statin usage and observed similar results (Supplementary Table 3).

## Impact of risk factors on total healthcare costs

To quantify the amount of annual total healthcare costs associated with WC, adult BMI, and SBP in absolute euro costs (instead of percent changes), we assumed a median annual total healthcare cost of €1312.53 (Table 1). One SD increase in WC, adult BMI, and SBP resulted in increases of €298.99, €179.03 and €171.68 of annual total healthcare costs, respectively. Using clinically interpretable units, we estimated €202.13 annual increase per additional 10 cm of WC; €178.51 per 5 kg/m^2 of adult BMI; and €84.00 increase per 10 mmHg of SBP. In addition to assuming a median annual total healthcare cost, we estimated the absolute euro costs at varying baseline healthcare costs for WC (Supplementary Fig. 3).

## Impact of risk factors on total healthcare costs by service type, sex, and age

We quantified the impact of six risk factors with significant effects on annual total healthcare costs by repeating the analyses by each service type (i.e., primary care, secondary care, medication), sex, and age (Fig. 4). SBP (medication vs. secondary care costs, $P = 5.75 \times 10^{-9}$ for difference in effect size) and triglycerides (medication vs. secondary care costs $P = 1.09 \times 10^{-4}$) had larger effects on medication costs than secondary (or primary) care costs. Such effects reflected relative rather than absolute increases. For example, a one SD increase in SBP caused a large relative difference in annual medication costs than secondary care costs (34.18%) increase (95% CI: [27.16, 41.59]) vs 8.17% increase (95% CI: [3.10, 13.49], respectively). However, the estimated absolute euro changes were similar (i.e., medication costs of €69.04 vs. secondary care costs of €69.61). Interestingly, a genetically-predicted SD increase in LDL cholesterol did not change primary or secondary care costs, but increased medication costs (8.07%) increase (95% CI: [3.54, 12.80], $P = 3.8 \times 10^{-4}$). Nevertheless, the prior result of the null effect of LDL cholesterol on annual total healthcare costs can be explained by the relatively lower magnitude of contribution of medication costs than secondary care costs.

We found little evidence that the relative impact of the risk factors on healthcare costs differ between females and males. Similarly, we found few differences between individuals younger than 30 years old, between 30 and 60 years old, and older than 60 years old. The only exception was a modest difference in the relative impact of SBP on healthcare costs between individuals aged 30 to 60 years old (7.79%, 95% CI: [2.12, 13.77]) compared to individuals older than 60 years old (18.38%, 95% CI: [13.71, 23.23]) for SBP ($P = 3.20 \times 10^{-3}$).

## Factors mediating the impact of risk factors on total healthcare costs

For the three risk factors with the largest percent change on healthcare costs (WC, adult BMI, SBP), we used MVMR to understand how much of their impact on healthcare costs can be explained by increased risk for major diseases associated with high healthcare costs (Supplementary Table 7). We considered the top five noncommunicable diseases from the Global Burden of Disease study:[14] back pain, chronic ischemic heart disease, type 2 diabetes, chronic obstructive pulmonary disease, and stroke. For SBP, we additionally studied blood pressure medications as a mediator, which was not immune from collider bias but provided context for indirect effects of SBP on healthcare costs.

After accounting for the genetic effects mediated by the five noncommunicable diseases, we found that type 2 diabetes and blood pressure medications modestly mediated the effects of adult BMI and SBP on annual total healthcare costs, respectively. Adjusting for type 2 diabetes slightly attenuated the effect of adult BMI on healthcare costs from 13.64% [95% CI: 10.26, 17.12] to 10.18% [95% CI: 4.88, 15.76]. Adjusting for blood pressure medications attenuated the effect of SBP on healthcare costs from 13.08% [95% CI: 8.84, 17.48] to 4.06% [95% CI: −2.45, 10.47]. Interestingly, even after adjusting for the top five noncommunicable diseases, WC effects on healthcare costs remained similar suggesting that WC affects healthcare costs broadly beyond the increased risk of the top five major diseases.

## Replication analysis for generalizability and robustness of healthcare costs findings

We conducted several analyses to evaluate the robustness of our findings. First, we perform similar MR analysis in UK Biobank ($N = 307,048$) and we estimated a £96.90 (€115.32) increase per SD of WC; a £94.59 (€112.57) increase per SD of adult BMI; and a £24.36 (€28.99) increase per SD of SBP. In clinical units, we estimated a £77.40 (€92.11) increase per 10 cm of WC; a £102.82 (€122.36) increase per 5 kg/m^2 of adult BMI; and a £11.77 (€14.01) increase per 10 mmHg of SBP. Similarly, in the Netherlands Twin Register ($N = 16,726$), we estimated a €182.52 increase per SD of WC; a €264.85 increase per SD of adult BMI; and a €10.07 increase per SD of SBP. In clinical units, we estimated a €129.91 increase per 10 cm of WC; a €261.44 increase per 5 kg/m^2 of adult BMI; and a €87.69 increase per 10 mmHg of SBP. Results from these other sources are therefore in the range of our estimates, despite the different healthcare systems, data sources (e.g., different cost categories captured), and population structures (Fig. 5, Supplementary Table 4).

Second, we compared the genetic association with annual healthcare costs in FinnGen with those publicly available from the United Kingdom and Netherlands (Supplementary Table 5). We observed that the genetic correlation was significant between between Finland, the United Kingdom, and Netherlands. Comparing secondary care costs, Finland and the United Kingdom had a genetic correlation of 0.804 (SE = 0.05492, $P = 1.61 \times 10^{-48}$). Comparing primary care costs, Finland and the Netherlands had a genetic correlation of 0.7694 (SE = 0.3387, $P = 2.31 \times 10^{-2}$). For the Netherlands total, secondary care, and medication costs, heritability was too low to calculate genetic correlation.

Third, we calculated the PGS for healthcare costs in FinnGen using weights from the United Kingdom (UK) and Netherlands (NL). In general, there was a large and significant association between Finnish healthcare costs and UK- and NL-based PGS, suggesting that cross-country analyses of healthcare costs may be valuable (Supplementary Figure 4). A 1 SD increase in the UK-based PGS for secondary care costs was associated with an increase in €128 per year (95% CI: [97, 160], $P = 2.09 \times 10^{-15}$) or 9.29% per year (95% CI: [8.74, 9.84], $P = 6.70 \times 10^{-293}$). A 1 SD increase in the NL-based PGS for total healthcare costs was associated with an increase of €15 per year (95% CI: [−19, 50],

**Table 1 | Monetary impact of three main biological risk factors for 343,160 FinnGen participants as estimated from Mendelian Randomization**

| Exposure | Percent change in cost per unit change in exposure | Confidence Interval | Estimated absolute change in euros |
|---|---|---|---|
| Waist circumference | 22.78 per 1 SD | [18.75, 26.95] | €298.99 per 1 SD |
| | 15.40 per 10 cm | [12.90, 17.90] | €202.13 per 10 cm |
| Body mass index | 13.64 per 1 SD | [10.26, 17.12] | €179.03 per 1 SD |
| | 13.58 per 5 kg/m^2 | [10.34, 16.84] | €178.51 per 5 kg/m^2 |
| Systolic blood pressure | 13.08 per 1 SD | [8.84, 17.48] | €171.68 per 1 SD |
| | 6.38 per 10 mmHg | [4.39, 8.37] | €84.00 per 10 mmHg |

Estimates for absolute euro costs are based on the median healthcare costs and may vary assuming different baseline healthcare costs. SD is standard deviation.

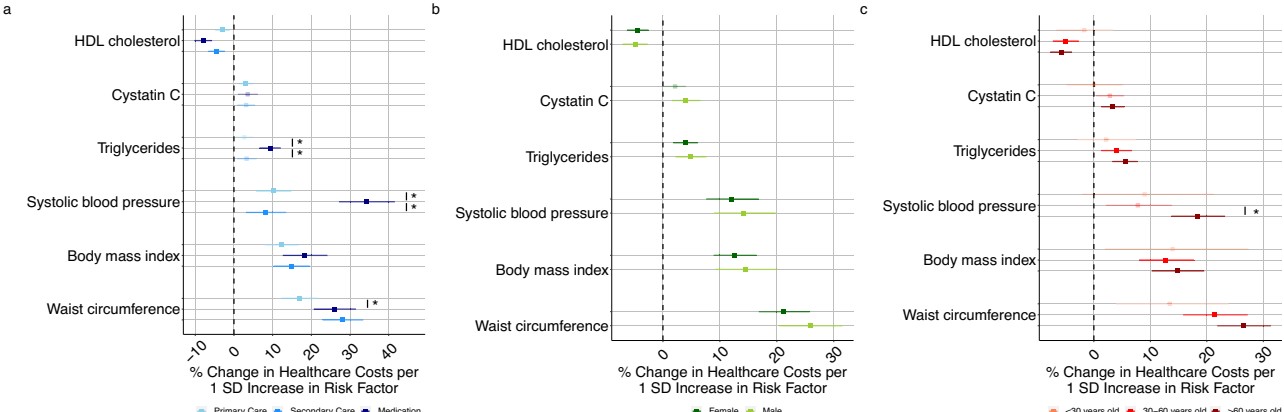

**Fig. 4 | Mendelian Randomization results for healthcare costs for six biological risk factors in FinnGen (N=373,160).** Mendelian Randomization on six biological risk factors for 373,160 FinnGen participants using the two-sample, inverse variance weighted approach stratified by (**a**) service type, (**b**) sex, and (**c**) age. There was a significant difference between the medication costs associated with triglycerides compared to primary care costs ($P = 1.09 \times 10^{-4}$) and secondary care costs ($P = 6.84 \times 10^{-4}$). There was a significant difference between the medication costs associated with systolic blood pressure compared to primary care costs ($P = 5.75 \times 10^{-9}$) and secondary care costs ($P = 2.29 \times 10^{-9}$). There was a significant difference between the medication costs associated with waist circumference

compared to primary care costs ($P = 6.81 \times 10^{-3}$). There was a significant difference in the total healthcare costs associated with systolic blood pressure for individuals older than 60 years old and individuals between 30 and 60 years old ($P = 3.21 \times 10^{-3}$). Two-sided p-values were calculated from the effect estimates and standard errors of the Mendelian Randomization model and adjusted for multiple hypothesis testing. Bars indicate 95% confidence interval. The *sign indicates significant differences between different levels of the stratification variable within the risk factor at the Bonferroni-corrected significance level ($P < 8.33 \times 10^{-3}$). SD is standard deviation. Source data are provided as a Source Data file.

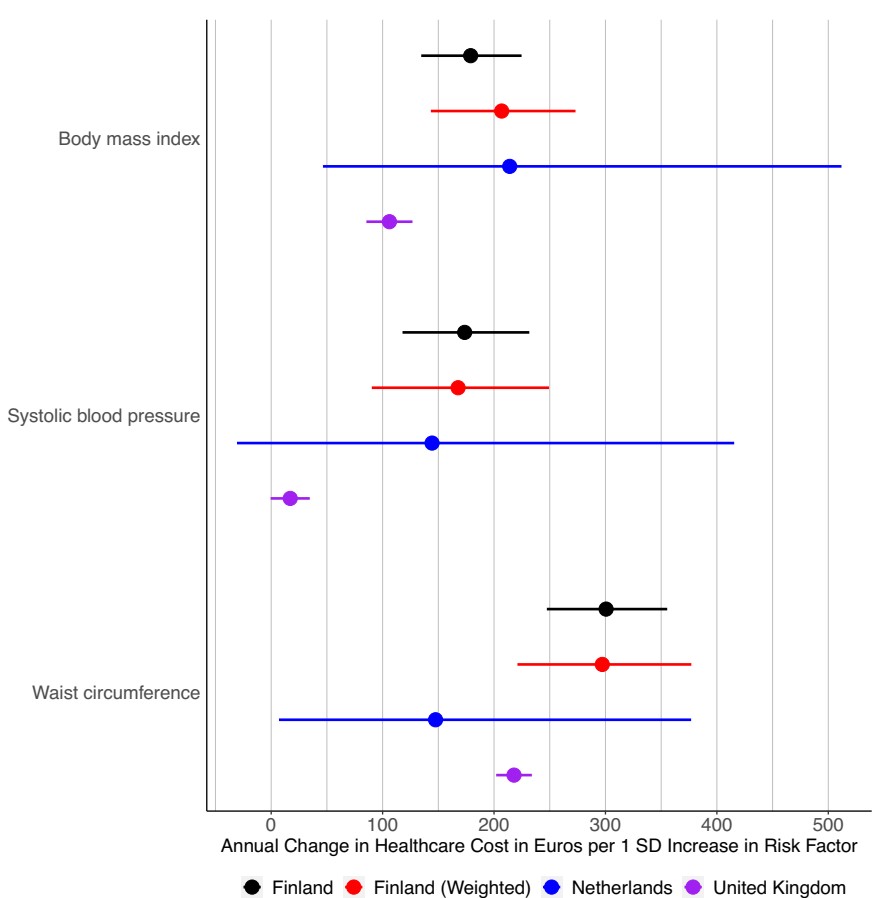

**Fig. 5 | Mendelian Randomization results for total healthcare costs for three main biological risk factors in a replication analysis including data from the United Kingdom (N = 307,048), Netherlands (N = 16,726), Finland (N = 373,160) and re-weighting the FinnGen cohort to reflect the entire Finnish population.**

Estimates for absolute euro costs are based on the median healthcare costs and may vary assuming different baseline healthcare costs. SD is standard deviation. Points indicate beta effect size estimates and bars indicate 95% confidence interval. Source data are provided as a Source Data file.

$P = 3.83 \times 10^{-1}$) or 2.84% per year (95% CI: [2.46, 3.21], $P = 1.66 \times 10^{-50}$). The lower increase observed for NL-based PGS is expected given the PGS was derived on a smaller sample size. Overall, despite lack of genome-wide significant signals in UK biobank and the Netherlands Twin Register, we could derive a PGS for healthcare costs with a significant effect in FinnGen suggesting consistency in genetic associations for healthcare costs across multiple countries.

Finally, FinnGen is not fully representative of the general Finnish population and enriched with individuals that have been in contact with the healthcare system due recruitment being predominantly based in hospital-based settings. To evaluate the generalizability of our results to the entire Finnish population, we used inverse probability weighting with weights calculated by comparing five health and sociodemographic characteristics (i.e., age, gender, education, occupation, and region of birth) between FinnGen participants and the full Finnish population (Supplementary Fig. 5, Supplementary Table 6). We found a high correlation between the effect sizes from the GWAS of healthcare costs and the weighted linear regression ($R^2 = 0.76$). We also found similar results for the MR analysis, in which one SD increase in WC increased healthcare costs by 22.64% (95% CI: [16.84, 28.72], $P = 1.53 \times 10^{-16}$), one SD increase in adult BMI increased healthcare costs by 12.42% (95% CI: [6.96, 18.16], $P = 4.06 \times 10^{-6}$), and one SD increase in SBP increased healthcare costs by 12.56% (95% CI: [6.66, 18.80], $P = 1.67 \times 10^{-5}$).

## Discussion

We linked genetic information to detailed healthcare costs covering primary, secondary, and medication costs for 373,160 participants in FinnGen followed-up to a maximum of 22 years. This allowed us to evaluate the association between the genetic underpinnings of 15 clinically relevant risk factors and annual total healthcare costs. Generally, making causal inferences about the effects of these risk factors is challenging because of confounding, reverse causation, and the unfeasibility of randomized controlled trials. We address these limitations using a genetically-informed causal inference design. Under the assumptions of Mendelian Randomization, we estimated the causal effects of these risk factors on total healthcare costs. Our approach was conservative, and we chose risk factors that have strong genetic bases and high heritability. However, we did not consider important modifiable risk factors such as smoking and alcohol consumption because using MR with such risk factors represents additional challenges.

The risk factor with the largest quantitative impacts on healthcare costs were WC, followed by adult BMI and SBP. An increase of 10 cm in WC results in 15.40% increase in annual healthcare costs, which, in Finland, corresponds to approximately €202.13. The effect of WC, unlikely BMI, was not attenuated when considering the potential mediating effect of five major diseases. Previous studies have suggested that WC may be more informative than adult BMI for certain health outcomes, as WC may better reflect the accumulation of intra-abdominal fat compared with BMI[15]. The MR study of Hazewinkel et al. in the UK Biobank found that an adverse fat distribution rather than the level of BMI may drive the relationship between BMI and higher rates of hospital admission[16].

Previous studies have used conventional approaches other than MR to quantify the effects of WC, adult BMI, and SBP on healthcare costs. Hojgaard et al[17]. used data from a Danish prospective cohort study and found that women with increased WC (> = 80 cm) incurred $261 more in annual healthcare costs than women with normal WC (< 80 cm) while men with increased WC incurred $420 more in annual healthcare costs than men with normal WC. Pendergast et al[18]. used data from an American and German prospective cohort study and found that individuals with higher WC have 16–18% (€300–€400) and 20–30% ($1900–$2400) higher healthcare costs compared to individuals with lower WC. Kirkland et al.[19] used data from a nationally

representative database from the United States and found that individuals with hypertension had $1920 higher annual healthcare costs than individuals without hypertension.

Importantly, we found that the impact several biomarkers (e.g., alanine aminotransferase, glycated hemoglobin, C-reactive protein, LDL cholesterol, lipoprotein(a), creatinine, vitamin D, albumin, and glucose) have on healthcare costs was modest and not significant at the Bonferroni-corrected significance level. It has been argued the MR is more valuable to reject causal claims when the genetic instrument is sufficiently strong[20], as in our case.

There may be two main reasons why we did not find significant effects for these biomarkers. First, elevated biomarkers can be consequences of underlying disease processes, for example, by reflecting inflammation, as in the case of C-reactive protein. Moreover, their levels can simply capture (un)healthy behaviors. For example, numerous trials have shown no benefits for Vitamin-D supplements on reducing risk for several diseases, such as cardiovascular diseases, despite supporting evidence from observational studies[21,22], but not from MR-based studies[23–25]. Second, the effect of risk factors on healthcare costs reflects current clinical practice. If a risk factor is routinely measured and those with high levels of the risk factor are correctly targeted by preventive interventions, the increased healthcare costs associated with the preventive interventions should be counterbalanced by the reduced healthcare costs associated with the prevented disease burden. Such is the case of LDL cholesterol—if LDL cholesterol was sufficiently treated in the population, LDL cholesterol would have less of an impact on healthcare costs, and we found a strong effect of LDL cholesterol on medication costs, but not primary and secondary care costs. Indeed, Harrison et al. observed a null impact of total serum cholesterol on other social and economic outcomes in the UK Biobank[26]. Similarly, while glycated hemoglobin is a known marker for type 1 and 2 diabetes, proper management may result in a lower impact of glycated hemoglobin on healthcare costs as compared to a theoretical scenario where patients with high glycated hemoglobin were untreated. Triangulation[27] with other evidence sources, such as randomized controlled trials, may better establish the mechanism through which medical services reduce specific categories of healthcare costs (e.g., whether preventative interventions indeed lower healthcare costs associated with downstream morbidity).

Our study has limitations. First, the power and precision of our MR analysis was limited by the availability of SNPs associated with the risk factors. Particularly, in our MVMR analysis, some mediators, such as stroke, had a small number of variants that likely affected statistical power. As more powerful GWAS are performed, MR may gain the statistical power to more accurately quantify causal effects. Second, our MR analysis was limited to linear instrumental variable estimates. The validity of the causal claims relies on the assumption that we are estimating the local average treatment effect (e.g., the average effect of a risk factor on healthcare costs for individuals whose exposure was affected by the value of the genetic instrumental variables)[28]. The linear instrumental variable estimates will be in the same direction as the causal effect for each individual in the population as long as the exposure-outcome relationship is monotonic. Thus, our MR analysis is valid in estimating the local average treatment effect of a risk factor on healthcare costs in our cohort rather than any non-linear effects between exposure and outcome for any single individual.

Third, MR uses the genetic variation assigned at conception (e.g., genetically determined risk factors), and therefore estimates the lifetime effects of risk factors on healthcare costs, rather than acute or temporary effects. For example, an intervention that reduces WC in older ages may not result in reductions in healthcare costs consistent with our estimates. Likewise, MR does not necessarily comply with the "stable unit treatment value" assumption of causal inference, as there may be different mechanisms of hypothetically manipulating an exposure. For example, an individual may have well-controlled SBP

because they were born with a favorable genotype or because they are taking blood pressure medications, and the impact of these differences influences on SBP may differ. Fourth, healthcare systems worldwide vary. Finland, which has a public healthcare system, is ideal for the analysis of healthcare costs, as healthcare services are uniformly priced, and is similar to other European countries with public healthcare systems. On the other hand, countries that rely more heavily on private healthcare and insurance, such as the United States, may offer healthcare services at different costs depending on insurance plans and other factors, making the analysis of healthcare costs difficult. Moreover, our results are based on individuals of European ancestry, and genetic effects might vary across ancestry groups. Fifth, while we comprehensively capture most healthcare costs in our cost estimates through using registry-based data, there are certain cost categories that are not captured by national healthcare registries (e.g., private occupational healthcare, institutional care for elderly and disabled individuals outside of a hospital setting, eyeglasses and other medical devices, and healthcare-related transportation costs such as ambulances and reimbursed taxis). As such, our cost estimates are representative of primary outpatient, secondary and tertiary inpatient and outpatient hospital visits, and medication costs.

Our approach might inform the cost-effectiveness of common healthcare screening procedures based on biomarkers measurement. More in general, linking genetics to healthcare costs opens different research venues. For example, evaluation of the costs associated with specific genetic variants that mimic drug targets may inform drug development and commercialization. Implementation of genetic screening either in the form of polygenic score or single variants, would require health-economic assessment[29–31]. Future large-scale genetic studies will be powered to provide a comprehensive assessment on the impact of genetics on healthcare costs and facilitate the implementation of such proposed genomic medicine approaches.

In conclusion, our results not only indicate that elevated WC, BMI and SBP are major causal contributors to healthcare costs, but could also quantify their impact on healthcare costs within a causal inference framework. This has implications for the cost-effectiveness of interventions and policies that influence these biomarkers. Several other biomarkers routinely measured in clinical setting are unlikely to directly impact on healthcare costs, either because they are not causal to healthcare cost, or because they are already well managed in the clinic.

## Methods
### Study cohort
This study utilized data from the FinnGen Study, which is an ongoing prospective cohort study aiming to recruit 520,000 individuals by combining population-based legacy cohorts, disease-based cohorts, and volunteers recruited by biobanks[32]. The average age at baseline (i.e., date of DNA sample collection) is 54 years old and 56% of the study cohort is female. Participants are linked to national health registries that provide rich longitudinal information. Such registries include the Register of Primary Health Care Visits (AvoHILMO) which captures outpatient visits, the Care Register for Health Care (HILMO) which captures hospital visits, and the Medication Reimbursement Register (Kela). Individual-level genotypes and register data from FinnGen participants can be accessed by approved researchers via the Fin-genious portal (https://site.fingenious.fi/en/) hosted by the Finnish Biobank Cooperative FinBB (https://finbb.fi/en/). Data release to FinBB is timed to the bi-annual public release of FG summary results which occurs twelve months after FG consortium members can start working with the data.

Given that the study participants in FinnGen may differ from the entire Finnish population due to its hospital-based recruitment (e.g., individuals in FinnGen are typically sicker and have higher disease prevalence), we adjusted the study cohort in FinnGen to the entire Finnish population using inverse probability weights in a subsequent sensitivity analysis. We used the calibration weighting method, which uses the marginal proportions of variables to adjust the sample weights to satisfy the population margins. We used the following five health and sociodemographic characteristics: age, gender, education, occupation, and region of birth.

Participants in FinnGen provided informed consent for biobank research on basis of the Finnish Biobank Act. The Coordinating Ethics Committee of the Hospital District of Helsinki and Uusimaa (HUS) approved the FinnGen study protocol (number HUS/990/2017). The FinnGen study is approved by the THL (approval number THL/2031/6.02.00/2017, amendments THL/1101/5.05.00/2017, THL/341/6.02.00/2018, THL/2222/6.02.00/2018, THL/283/6.02.00/2019 and THL/1721/5.05.00/2019), the Digital and Population Data Service Agency (VRK43431/2017-3, VRK/6909/2018-3 and VRK/4415/2019-3), the Social Insurance Institution (KELA) (KELA 58/522/2017, KELA 131/522/2018, KELA 70/522/2019 and KELA 98/522/2019) and Statistics Finland (TK-53-1041-17).

### Estimation of healthcare costs
(1) AvoHILMO and (2) HILMO are registries maintained by the Finnish Institute for Health and Welfare (THL) for (1) primary outpatient and (2) secondary and tertiary inpatient and outpatient hospital visits, respectively. The Finnish Institute for Health and Welfare publishes average unit cost estimates for different types of healthcare services (e.g., outpatient visits, inpatient episodes). The Social Insurance Institution (SII, also known as Kela), the Finnish government agency in charge of national social security programs, maintains a registry of all reimbursed prescription medication purchases in Finland. The Avo-HILMO registry was started in 2011, the HILMO registry in 1998, and the medication purchases registry in 1998. All AvoHILMO, HILMO, and medication costs capture total costs regardless of the payer. We did not capture costs related to private occupational healthcare, institutional care for elderly and disabled individuals outside of a hospital setting, eyeglasses and other medical devices, and healthcare-related transportation costs such as ambulances and reimbursed taxis. Going forward, we referred to any AvoHILMO costs as "primary care costs", HILMO costs as "secondary care costs", and Kela costs as "medication costs".

We used the unit cost estimates published by the Finnish Institute for Health and Welfare to obtain costs associated to each medical encounter[33]. Primary care costs were linked to each medical encounter by profession (e.g., physician, nurse), service type (e.g., primary healthcare, mental health), and contact type (e.g., visit, phone call). Secondary care costs were linked based on service (e.g., emergency room visit, outpatient visit, inpatient visit), specialty (e.g., cardiology, neurology), and hospital (e.g., university, central, other) types. Medication costs were linked using the Nordic Article Number (VNR), which is an identifier that exactly captures the type of medicinal product (e.g., manufacturer, dosage) purchased. We used the yearly average costs for each VNR code across Finnish pharmacies to link the costs. Primary care costs prior to 2011 were excluded, and secondary care and medications costs prior to 1998 were excluded to reflect the start dates of each registry. Individuals with secondary care or medication records, but without primary care records, were assumed to be individuals using private primary healthcare services. There were 294 (0.08%) such individuals in FinnGen, and they were assigned the median primary care cost of €71.62. For all cost categories, we examine costs in 2017 euro values such that the same service contributes similarly to costs whether it occurred for example in 2010 or 2017. Other missing values were assigned zero values (i.e., individuals with primary and secondary care records, but without medication records, were assigned a zero value for medication costs). To adjust for fluctuating healthcare costs by different years, each unique set of identifiers was assigned to the standardized healthcare costs in 2017.

We estimated the annual total healthcare costs, primary care, secondary care, and medication costs adjusted by the total follow-up time that individuals were observed in each registry. The start of follow-up was defined as 2011 for AvoHILMO and 1998 for HILMO and Kela. The end of follow-up (EOF) was defined as date of death, date of emigration, or the end-of-registry date (October 11, 2021). The annual total healthcare costs for each individual are estimated as:

$$
\begin{aligned}
Annual\ total\ healthcare\ costs = & \frac{Total\ primary\ care\ costs}{Total\ person-years\ from\ 2011\ to\ EOF} \\
& + \frac{Total\ secondary\ care\ costs}{Total\ person-years\ from\ 1998\ to\ EOF} \quad (1) \\
& + \frac{Total\ medication\ costs}{Total\ person-years\ from\ 1998\ to\ EOF}
\end{aligned}
$$

As healthcare costs were highly right-skewed in this sample, a $log(X+1)$ transformation was implemented before modeling. Log transformation is frequently applied to normalize right-skewed healthcare costs, although certain limitations remain (e.g., handling zero cost estimates and requiring an adjustment such as $log(X+1))$[34–37]. Log transformation was also used to make effect estimates more interpretable as effect estimates calculated from log-transformed outcomes yields percent changes (e.g., one standard deviation increase in the dependent variable yields a certain percent change in the outcome). Percent changes, rather than raw euro values, may be more interpretable as different countries utilize different currency systems, have different magnitudes of healthcare expenditure, etc. Annual total healthcare costs were the main outcome studied. In sensitivity analyses, we examined healthcare costs stratified by: (1) service type (e.g., primary care, secondary care, and medication costs), (2) sex, and (3) age at the end of follow-up (individuals under 30 years old, individuals between 30 and 60 years old, and individual over 60 years old).

### Genome-wide association study

The primary outcome was log-transformed annual total healthcare costs, and our secondary outcomes included (1) log-transformed primary care, secondary care, and medication costs, (2) log-transformed annual total healthcare costs for females and males, and (3) log-transformed annual total healthcare costs for individuals under 30 years old, individuals between 30 and 60 years old, and individuals over 60 years old. We performed genome-wide association studies (GWAS) of healthcare costs to identify genetic variants associated with healthcare costs using REGENIE, which is a method for fitting a whole-genome regression model[38]. Briefly, REGENIE uses a two-step process that fits a whole-genome regression model and performs single-variant association testing. We used the default model with the following covariates: birth year, birth year squared, sex, 10 principal components, and batch covariates. SNPs were filtered using MAF > 0.001 and INFO > 0.8.

### Mendelian Randomization

We performed MR, which is a method that uses genetic variants as instrumental variables to estimate the effect of specific exposures on healthcare costs[9]. The exposures included 15 biological risk factors based on the following criteria: (1) has strong genetic instruments (e.g., F-statistic > 50) and (2) of clinical interest and relevance (e.g., can be measured through available laboratory tests). We used summary statistics from the GWAS of healthcare costs conducted in FinnGen for the outcomes and non-overlapping summary statistics from the MRC IEU OpenGWAS Database for the exposures (Supplementary Table 1)[39]. Some summary statistics were back-transformed from standardized to raw units on the original scale (e.g., adult body mass index, HDL cholesterol, LDL cholesterol, triglycerides, systolic blood pressure, and waist circumference).

To evaluate the causal effect of the 15 risk factors on healthcare costs, we performed two-sample MR, which utilizes summary statistics from GWAS of exposures and outcomes in non-overlapping cohorts[40]. MR relies on several assumptions: (1) genetic instruments must be robustly associated with the exposure, (2) there must be no confounders of the genetic instruments-cost associations, and (3) genetic instruments must not influence costs except through the exposure of interest[9]. We performed two-sample MR using the TwoSampleMR package version 0.5.6 in R version 4.1[39,41].

We performed LD clumping with a window of 10000 kilobases and an $R^2$ cutoff of 0.002 and utilized the MR Egger, weighted median, inverse variance weighted, simple mode, and weighted mode methods. The inverse variance weighted method estimates the causal effect based on a ratio of association estimates from a univariable regression of the outcome on the genetic variant and the exposure on the genetic variant, averaging each ratio estimate with inverse variance weights. The MR Egger method uses a similar method, with the inclusion of an intercept term. The simple mode, weighted mode, and weighted median methods rely on similar approaches, with different weights. Several methods were used in combination due to differing advantages and disadvantages. For example, the MR Egger method, is more robust to pleiotropy (e.g., one variant affecting multiple phenotypes), yet suffers from lack of power, while the inverse variance weighted method retains more statistical power.

### Multivariable Mendelian randomization

Multivariable MR (MVMR) uses genetic variants for two or more exposures to simultaneously estimate the causal effect of each exposure on the outcome, controlling for the effect of the other included exposures. MVMR can therefore use genetic variants for several risk factors to estimate independent and direct effects of these risk factors, as well as estimating mediation[42]. MVMR requires the same assumptions as univariate MR, but the genetic instruments must be associated with the set of exposures rather than the single exposures, but it is not necessary for each genetic instrument to be associated with every exposure[42].

We performed MVMR to identify mediators of the exposures on healthcare costs, in which the mediators were the top five non-communicable diseases from the Global Burden of Disease:[14] back pain, chronic ischemic heart disease, type 2 diabetes, chronic obstructive pulmonary disease, and stroke. Summary statistics for mediators were obtained from the UK Biobank (Supplementary Table 1). For example, we estimated the simultaneous effects of chronic ischemic heart disease and waist circumference on healthcare costs by including two exposures in our model, which were identified using genetic variants associated with each exposure. This allowed us to obtain the direct effect of waist circumference on healthcare costs by adjusting out the indirect effect of chronic ischemic heart disease on healthcare costs.

### Replication analyses in the United Kingdom and Netherlands

We performed validation analyses in the UK Biobank[10] ($N = 307,048$) and Netherlands Twin Register[43] ($N = 16,726$) to evaluate the generalizability and robustness of our results to different healthcare systems in different countries. For the UK Biobank, annual total healthcare costs were used as the primary outcome, in which pounds were converted to euros using the average exchange rate in 2021 of 1 EUR = 0.8403 GBP, and in-house GWAS protocols for quality control were used[44]. Briefly, the following exclusion criteria were applied: (1) individuals with mismatches between their biological sex and self-reported gender, (2) individuals with sex chromosome aneuploidy, and (3) related individuals of white European British ancestry[45]. The analysis was restricted to autosomal variants using a graded filtering process to account for imputation quality at different allele frequency ranges so that rarer genetic variants were required to have a higher imputation INFO score. Linear mixed models were implemented using BOLT-LMM and controlled for age, sex, and the first

40 principal components[46]. For the Netherlands Twin Register, we used published GWAS summary statistics[43].

We repeated our analyses to estimate the annual monetary impact per capita associated with each risk factor. We applied two-sample MR using largely the same summary statistics for risk factors and summary statistics from the UK Biobank[10] and Netherlands Twin Register[43] for healthcare costs. For the risk factor summary statistics that included individuals from the UK Biobank, we used separate summary statistics to ensure non-overlapping summary statistics between exposures and outcomes (Supplementary Table 1). We also calculated the genetic correlation between the three sets of summary statistics (e.g., GWAS on healthcare costs in Finland, United Kingdom, and Netherlands) to evaluate the consistency of associations across different healthcare systems using linkage disequilibrium score regression (LDSC), a tool to estimate genetic correlation and heritability[47]. Briefly, after filtering (e.g., INFO > 0.9 and minor allele frequency (MAF) > 0.01), LDSC with default parameters and Hapmap3 SNPs in the 1000 Genomes Project European reference panel was used[47].

We also constructed polygenic scores (PGS) of healthcare costs from the UK Biobank and Netherlands Twin Register and estimated their associations with healthcare costs in FinnGen. Using the GWAS summary statistics from the UK Biobank and Netherlands Twin Register to construct PGS in the FinnGen cohort provided an additional approach on top of genetic correlation to understand if genetic signals in one study could replicate in the other studies, despite difference in the healthcare costs definitions across countries. We first used PRS-CS[48] to calculate weights of association and PLINK2[49] to calculate scores. Briefly, PRS-CS is a polygenic prediction method that uses Bayesian regression and infers posterior SNP effect sizes under continuous shrinkage priors using only GWAS summary statistics and an external linkage disequilibrium reference panel[48]. The 1000 Genomes reference panel was used to output weights using standard PRS-CS parameters. Only HapMap3 variants were included. PLINK2 was used to calculate the PGS, which were then standardized across the entire FinnGen cohort with a mean of 0 and a standard deviation of 1.

### Reporting summary

Further information on research design is available in the Nature Portfolio Reporting Summary linked to this article.

## Data availability

All data are made available from the FinnGen Study with research proposals approved by institutional review board and the FinnGen Scientific Committee. All individual-level data from the FinnGen Study is under controlled access due to the sensitive nature of health information data and can be requested from the Finnish Biobank Cooperative FinBB (https://site.fingenious.fi/en/). Other health register data can be requested from the Finnish Data Authority Findata (https://findata.fi/en/permits/). All Finnish biobanks can provide access for studies under the scope of the Finnish Biobank Act, broadly using biobank samples or data to promote health, understanding disease mechanism, or developing interventions used in healthcare. Supporting genome-wide association data supporting the findings of this study are publicly available on request from the UK Biobank (https://www.ukbiobank.ac.uk/) and the Netherlands Twin Register (https://www.nimhgenetics.org/download-tool/NTR). All source data are provided with this paper. All data supporting the findings of the study are available in the article, in the Supplementary Information, or from the corresponding author upon request. Source data are provided with this paper.

## Code availability

All code is available via Github: https://github.com/jiwooleebroadinstitute/healthcare_cost_finngen (https://doi.org/10.5281/zenodo.8184260).

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

## Acknowledgements

We would like to acknowledge Camiel M. van der Laan and Dorret I. Boomsma for their contribution of healthcare costs summary statistics from the Netherlands Twin Register, as well as Bart Ferket for his insights on health economics in designing this study. We would also like to acknowledge all of the study participants for their generous participation in FinnGen and other biobanks, as well as FinnGen as a study group that has contributed to this study.

A.G. was supported by the Academy of Finland (grant no. 323116) and by the European Research Council under the European Union's Horizon 2020 Research and Innovation Programme (grant no. 945733). This project has also received funding from the European Union's Horizon 2020 Research and Innovation Programme under grant agreement no. 101016775. The Medical Research Council (MRC) and the University of Bristol support the MRC Integrative Epidemiology Unit [MC_UU_00011/1]. NMD was supported via a Norwegian Research Council grant number 295989.

## Author contributions

A.G. and J.L. designed the study. J.L. calculated healthcare costs, performed the genome-wide association study, produced the Mendelian randomization results, and generated all figures and tables. S.J. contributed to calculating healthcare costs in FinnGen. A.K. contributed to calculating prescriptions costs in FinnGen. P.D. and N.D. contributed to producing the Mendelian randomization results in the UK Biobank. G.D.S., P.D., and A.G. contributed to the interpretation of the results. J.L. wrote the manuscript with feedback from all co-authors.

## Competing interests

P.N. reports research grants from Allelica, Apple, Amgen, Boston Scientific, Genentech / Roche, and Novartis, personal fees from Allelica, Apple, AstraZeneca, Blackstone Life Sciences, Foresite Labs, Genentech / Roche, GV, HeartFlow, Magnet Biomedicine, and Novartis, scientific advisory board membership of Esperion Therapeutics, Preciseli, and TenSixteen Bio, scientific co-founder of TenSixteen Bio, equity in MyOme, Preciseli, and TenSixteen Bio, and spousal employment at Vertex Pharmaceuticals, all unrelated to the present work.
