## [Peer Review File · Nature Communications]

nature portfolio

Peer Review FileReviewer comments, first round

Reviewer #1 (Remarks to the Author):

The work from Lee et al uses Mendelian Randomization to evaluate the economic burden of common risk factors on health care costs. In this respect it is both innovative and clearly deals with an important topic.

Overall the paper is clear and I think the conclusions are in line with the results the authors obtain.

I just have a few comments regarding mostly the presentation and the interpretation of the data.

Why was the 1000 Genomes used as reference LD dataset if UK biobank (which is the population where the GWAS was performed) was available? I realise that this will likely not affect the results deeply (although it may change the r^2), yet it would have been more correct.

For figure 2 it would have been better to show the density rather than the total number of individuals when comparing across different groupings. Given that the category numerosity it is not always so clear if there is a difference in distribution. This is evident mostly in Female vs Male comparison where apparently men have a lower cost but they are also only 44% of the sample.

Why have the GWAS results been omitted from the results? It would have been interesting to see if any of the top genes overlap with genes known to be associated to exposures (ie FTO).

I think the interpretation of the authors regarding the lack of a significant effect of LDL on total health care cost is correct. I wonder if stratifying by type show some difference finding a significant effect on medication? Something similar seems to be happening for SBP where the large effect is placed on medication.

Why is the difference in healthcare cost referred to in terms of percentage difference rather than in absolute terms?

Another important point regarding the scale is in my opinion the use of the $\log(x+1)$ to transform the data. This means that the relationship with the exposure is not linear but it will increase differently depending on where on the curve the exposure measure is. So a decrease within normal range at the centre of the exposure distribution will impact the expenditure less than at the right tail. This actually makes sense and it would be nice to see what the predicted expenditure would look like across different values of the exposure for example in a plot.

This impacts particularly table 1 and the previous paragraph in terms of interpretation which is presented as constant.

Minor comments.

Line 210 states “The MR Egger method uses a similar method, with the exclusion of an intercept term” I should actually be “inclusion of an intercept term”.

Reviewer #2 (Remarks to the Author):

Lee and colleagues were to quantify the causal impact of 15 biological risk factors on healthcare costs using data from 373,160 participants from the FinnGen Study and Mendelian Randomization (MR) approaches. The study found that higher waist circumference (WC), higher body mass index (BMI), and higher systolic blood pressure (SBP) consistently led to a higher health care cost. Results for other 12 risk factors are mixed, 3 additional risk factors lead to a higher cost in some components of the cost, the rest 9 risk factors led to no changes in health care costs.

I will leave biostatisticians to comment on the appropriateness of using MR to study health care costs and if this method was executed correctly as I am not an expert on MR and its application in generic research. My comment below was based on my thinking as an economist.

1. Linking risk factors to health care cost or quantifying the contribution of a risk factor is a complex issue. How much a person spends on health care is determined by many factors such as genetic factors, health behaviors, health care finance system, patient ability to pay, health/disease conditions in particular. Among those factors, genetic variations may explain only a very small part of the health care spending. As the authors did not control many factors that are important factors to explain high health care costs, it is hard to know if result may change while more factors were added to the model. The author control only top five conditions based on global disease burden. There are many health conditions which have large effects on health care cost such as cancer, kidney disease, HIV.. Results may well change when more diseases or other factors that affect health care cost such as were controlled for.

2. Economists have been using genetic variations as instrumental variables to control econometric problems such as simultaneity, confounding...while quantifying the impact of a risk factor on health care expenditure. Presumption for this practice is these risk factors are associated with a high health care cost as they are risk factors for diseases. Generic risk factors

were used as instrument variables for solving econometric issue not to study whether the risk factors lead to higher health care costs. In other words, risk factors always contribute to a higher health care cost. A risk factor was not found lead to a higher health care cost could means other things other than the risk does not lead a higher health care cost such as not a good instrument for the risk variable, or inadequate control, data problem.....

3. The author used average health care cost of many years and risk factors at a given time. However, many risk factors such as WC, BMI, Blood pressure., glucose level, Hb A1c.....can change over time due to lifestyle change or clinical treatments. Changes in risk factors created a complex relationship between a risk factors and health care costs. I am not sure using an average health care cost of many years and an observed one-time risk factor in a regression model is appropriate.

4. Some of the results did not make intuitive sense. For example, diabetes is one of the most expensive diseases. Diabetes is defined as one who had a high a blood glucose level. It is hard to believe that high glucose or HbA1c would not lead to high health care cost.

Reviewer #3 (Remarks to the Author):

The authors performed Mendelian Randomization (MR) to estimate the effects of 15 risk factors on annual total healthcare costs, using data from FinnGen (N=373,160), UK Biobank (N=307,048), and Netherlands Twin Register (N=16,726). Multiple MR methods were used to estimate the effects and stratified analyses (by service type, age, and sex) were performed. Multivariable MR analysis was also conducted to estimate the effect of risk factors on healthcare cost adjusted for major diseases. The major findings the authors showed were 1) significant effects of waist circumference, adult body mass index, and systolic blood pressure on increased annual total healthcare costs, 2) differences and consistency of these effects by service type, sex, and age, and 3) replication of these findings across Finland, UK, and Netherlands.

The methods in the study are generally sound and robust, and the MR findings are novel and of interest to not only the human genetics field but also sociogenomics and public health. I have only a couple of minor comments:

1. The description of running GWAS of healthcare costs in the UK Biobank seems to be missing? I could not find a description of how that GWAS was performed and the GWAS was not part of the original study by Dixon et al. Please highlight or add a description of the samples and methods used for the GWAS of healthcare costs in the UK Biobank.
2. I understand the authors do not want to focus on the GWAS of healthcare costs but the MR analysis utilizing those GWAS. However, a separate section in the Methods for GWAS of healthcare costs in FinnGen (and for UK Biobank) is more appropriate than embedding it in

the MR section.

3. The quality of the GWAS of healthcare costs is not described anywhere in the paper. I could not find any description of basic GWAS summary statistics metrics such as lambdaGC or LD score regression intercept, significant association findings, Manhattan plot, Q-Q plot etc. It would be very helpful to provide the information for the reader to judge the quality of the GWAS (as a source data of the MR analyses).
4. The GWAS of healthcare costs are by themselves of interest to many fields as I mentioned above. I would suggest the authors to share the summary statistics publicly to facilitate the research fields (if this is not requested by the journal already).
5. I would encourage the authors to discuss the current findings of effects of the risk factors on healthcare costs with previous literatures of the same topics but based on conventional epidemiological approaches, or at least acknowledge previous studies on the same topic (but with different approaches) exist (or not?)
6. There is a very minor conceptual point on the section title “Factors mediating the impact of risk factors on total healthcare costs”. The MVMR analysis described in this section does not provide an estimate of “the impact of risk factors on total healthcare costs” mediated through other major diseases (as suggested by the section title). MVMR provides estimates of risk factors’ effects on total healthcare costs adjusted/controlling for other major diseases. The text in this section actually used terms such as “accounting” and “adjusting”, not mediating. The authors may consider change the section title.
7. Information on blood pressure medications GWAS and instrument is missing?
8. For the MVMR analysis, the authors should consider (in the Discussion) that the imbalance of number of instruments (ranged from 1 to 52; ST1) for the five major diseases may affect the MVMR analysis results. The only attenuation effect observed were for BMI’s effect when adjusting for type 2 diabetes and for SBP when adjusting for blood pressure medications. This could just be a reflection of lack of power for the other diseases due to few instruments.

REVIEWER COMMENTS

Reviewer #1 (Remarks to the Author):

The work from Lee et al uses Mendellian Randomization to evaluate the economic burden of common risk factors on health care costs. In this respect it is both innovative and clearly deals with an important topic.

Overall the paper is clear and I think the conclusions are in line with the results the authors obtain.

I just have a few comments regarding mostly the presentation and the interpretation of the data.

Comment 1: Why was the 1000 Genomes used as reference LD dataset if UK biobank (which is the population where the GWAS was performed) was available? I realise that this will likely not affect the results deeply (although it may change the r^2), yet it would have been more correct.

Answer: The summary statistics for our exposures came from a variety of GWAS including the UK Biobank and other sources, so the 1000 Genomes reference panel was used as a more broadly applicable reference panel, for example to identify independent loci. Furthermore, our in-house pipelines use the 1000 Genomes reference panel in generating polygenic risk scores, and when constructing these scores on the FinnGen cohort using summary statistics from two European cohorts (e.g., the UK Biobank and the Netherlands Twin Register), the same default PRS protocols were used.

Comment 2: For figure 2 it would have been better to show the density rather than the total number of individuals when comparing across different groupings. Given that the category numerosity it is not always so clear if there is a difference in distribution. This is evident mostly in Female vs Male comparison where apparently men have a lower cost but they are also only 44% of the sample.

Answer: In figure 2, we have now overlaid the histogram with the density plot to better represent the data.

Comment 3: Why have the GWAS results been omitted from the results? It would have been interesting to see if any of the top genes overlap with genes known to be associated to exposures (ie FTO).

Answer: We thank the reviewer for this suggestion to add more information about the GWAS results and we plan on making the summary statistics publicly available via the GWAS Catalog. Nonetheless, the focus of this paper is on the Mendelian Randomization analysis, and we have commenced related work, in collaboration with other studies, on a GWAS meta-analysis of healthcare costs for which full results and summary statistics will be made available. We have added the following text to the manuscript on page 10:

GWAS performed on log-transformed annual total healthcare costs identified several genome-wide significant SNPs (Supplementary Table 2 and Supplementary Figure 1 and 2).

We have also added a supplementary figure for the Manhattan plot, Q-Q plot, and a supplementary table for the top genome-wide significant hits (truncated for visibility).

Supplementary Figure 1. Manhattan plot from GWAS of log-transformed annual total healthcare costs in FinnGen.

Comment 4: I think the interpretation of the authors regarding the lack of a significant effect of LDL on total health care cost is correct. I wonder if stratifying by type show some difference finding a significant effect on medication? Something similar seems to be happening for SBP where the large effect is placed on medication.

Answer: We thank the reviewer for pointing out this result. Indeed, we noticed that LDL cholesterol, like systolic blood pressure, had a strong signal driven by medication costs. While a standard deviation in LDL cholesterol had no significant change on primary and secondary care costs, it increased medication costs by 8.07% (95% CI: [3.54, 12.80], $P = 3.8 \times 10^{-4}$). This is consistent with our thought that LDL cholesterol does not have a significant effect on annual total healthcare costs because it is sufficiently treated in the population (e.g., the medication costs counterbalance the primary and secondary healthcare costs associated with prevented disease burden). The following has been added to the manuscript on page 13 and 19:

Interestingly, a genetically-predicted SD increase in LDL cholesterol did not change primary or secondary care costs, but increased medication costs (8.07% increase (95% CI: [3.54, 12.80], $P = 3.8 \times 10^{-4}$). Nevertheless, the prior result of the null effect of LDL cholesterol on annual total healthcare costs can be explained by the relatively lower magnitude of contribution of medication costs than secondary care costs.

Such is the case of LDL cholesterol - if LDL cholesterol was sufficiently treated in the population, LDL cholesterol would have less of an impact on healthcare costs, and we found a strong effect of LDL cholesterol on medication costs, but not primary and secondary care costs.

Comment 5: Why is the difference in healthcare cost referred to in terms of percentage difference rather than in absolute terms?

Answer: We chose to report the difference in healthcare costs in terms of percentage difference because we used log-transformed healthcare costs and back-transforming the values resulted in the interpretation in terms of percentage difference. The percentage differences also allows us to account for differences in magnitudes between types of cost (e.g., medication costs are generally lower in magnitude overall than secondary care costs) and to make comparisons across different nations (e.g., Finland vs. United Kingdom vs. Netherlands) more interpretable. We do, however, report the difference in healthcare costs in absolute terms for our primary results in Table 1 for waist circumference, body mass index, and systolic blood pressure for more interpretability.

Comment 5: Another important point regarding the scale is in my opinion the use of the $\log(x+1)$ to transform the data. This means that the relationship with the exposure is not linear but it will increase differently depending on where on the curve the exposure measure is. So a decrease within normal range at the centre of the exposure distribution will impact the expenditure less than at the right tail. This actually makes sense and it would be nice to see what the predicted expenditure would look like across different values of the exposure for example in a plot.

This impacts particularly table 1 and the previous paragraph in terms of interpretation which is presented as constant.

Answer: We thank the reviewer for this comment. Using the $\log(x + 1)$ transformation we can interpret the estimates as *the percent change in healthcare costs per one standard deviation increase in the exposure*. The relationships between the change in one standard deviation and the corresponding percent change is linear. However, as the reviewer notes, Table 1 reports the absolute change in Euro based on the median annual healthcare costs, and the absolute change in healthcare costs will change at the extreme ends of the distribution. To provide context for this interpretation, we have added the following text to the manuscript on page 12:

In addition to assuming a median annual total healthcare cost, we estimated the absolute euro costs at varying baseline healthcare costs for WC (Supplementary Figure 3).

We have also improved the legends for Table 1 and Figure 5:

Table 1. Monetary impact of three main biological risk factors for 343,160 FinnGen participants as estimated from Mendelian Randomization. Estimates for absolute euro costs are based on the median healthcare costs and may vary assuming different baseline healthcare costs. SD is standard deviation.

Figure 5. Mendelian Randomization results for total healthcare costs for 3 three main biological risk factors in a replication analysis including data from the United Kingdom (N=307,048), Netherlands (N=16,726), Finland (N=373,160) and re-weighting the FinnGen cohort to reflect the entire Finnish population. Estimates for absolute euro costs are based on the median healthcare costs and may vary assuming different baseline healthcare costs. SD is standard deviation.

We have added a new supplementary figure showing the estimated the absolute euro costs as a function of baseline healthcare costs for waist circumference.

Supplementary Figure 3. Monetary impact of waist circumference for 343,160 FinnGen participants as estimated from Mendelian Randomization at varying baseline healthcare costs. SD is standard deviation.

Minor comments.

Comment 6: Line 210 states “The MR Egger method uses a similar method, with the exclusion of an intercept term” I should actually be “inclusion of an intercept term”.

Answer: We thank the reviewer for pointing out this typo. The following has been modified in the manuscript on page 7:

The MR Egger method uses a similar method, with the inclusion of an intercept term.

Reviewer #2 (Remarks to the Author):

Lee and colleagues were to quantify the causal impact of 15 biological risk factors on healthcare costs using data from 373,160 participants from the FinnGen Study and Mendelian Randomization (MR) approaches. The study found that higher waist circumference (WC), higher body mass index (BMI), and higher systolic blood pressure (SBP) consistently led to a higher health care cost. Results for other 12 risk factors are mixed, 3 additional risk factors lead to a higher cost in some components of the cost, the rest 9 risk factors led to no changes in health care costs.

I will leave biostatisticians to comment on the appropriateness of using MR to study health care costs and if this method was executed correctly as I am not an expert on MR and its application in generic research. My comment below was based on my thinking as an economist.

Comment 1. Linking risk factors to health care cost or quantifying the contribution of a risk factor is a complex issue. How much a person spends on health care is determined by many factors such as genetic factors, health behaviors, health care finance system, patient ability to pay, health/disease conditions in particular. Among those factors, genetic variations may explain only a very small part of the health care spending. As the authors did not control many factors that are important factors to explain high health care costs, it is hard to know if result may change while more factors were added to the model. The author control only top five conditions based on global disease burden. There are many health conditions which have large effects on health care cost such as cancer, kidney disease, HIV.. Results may well change when more diseases or other factors that affect health care cost such as were controlled for.

Answer: We thank the reviewer for this comment. We agree that many factors influence healthcare spending. Our focus on genetic factors is not because we expect genetic factors to be the most important in determining these outcomes. Instead, we use quasi-random variation in the allocation of genetic variants that influence liability to different exposures (e.g., the health conditions and risk factors) as instrumental variables to identify the causal effect of these exposures on healthcare spending. Put differently, we use this quasi-random genetic variation to econometrically identify these effects. The key novelty of this work is that we implemented the use of these genetics factors as instrumental variables to understand the causal impact of biological risk factors on healthcare costs. While we use genetic factors as instrumental variables, the focus of our work is on the relationships between the biological risk factors and the healthcare outcomes, not on the genetic factors themselves. Given this identification, we can answer causal questions, but do not claim that genetics is as important as other influences on healthcare spending.

The issue of controlling for other variables influencing healthcare cost does not arise in this type of instrumental variable study design, except to the extent that it is necessary to control for potential biases associated with the use of genetic variants as instrumental variables. Since these genetic variants are assigned at conception, any post-conception variables are very likely to constitute "bad controls" in the sense that they are potentially outcomes of the exposure itself. In other words, using the instrumental variable design of Mendelian Randomization, all other possible contributors to healthcare costs are downstream consequences of changes in the

biological risk factors of interest - these intermediate factors on the causal path from exposure to outcome therefore do not need to be adjusted for. We do however control for variables such as population structure (using genetic principal components to account for differential patterns of ancestral migration), as well as age and sex to account for potential influences on our outcome that are not downstream of the specific instruments we use.

Comment 2: Economists have been using genetic variations as instrumental variables to control econometric problems such as simultaneity, confounding...while quantifying the impact of a risk factor on health care expenditure. Presumption for this practice is these risk factors are associated with a high health care cost as they are risk factors for diseases. Generic risk factors were used as instrument variables for solving econometric issue not to study whether the risk factors lead to higher health care costs. In other words, risk factors always contribute to a higher health care cost. A risk factor was not found lead to a higher health care cost could means other things other than the risk does not lead a higher health care cost such as not a good instrument for the risk variable, or inadequate control, data problem.....

Answer: We thank the reviewer for this comment. We consider that the principal presumption for the analyses was not necessarily to confirm a pre-existing hypothesis that these risk factors were known to associate with high healthcare cost. Instead, these analyses typically proceed by examining whether such an association exists and then subsequently to estimate the direction and scale of any such causal effect. In other words, while several biological risk factors might be observationally associated with healthcare costs, this does not mean that they are causally associated with healthcare cost. Take, for example, C-reactive protein (CRP), which is a marker of inflammation. CRP has been associated with cardiovascular diseases and other diseases using observational data [1]. Thus, one would expect higher level of CRP to be associated with healthcare costs. However, CRP is a marker of inflammation and inflammation might be the consequence of underlying disease processes [2]. Higher genetically instrumented CRP is unlikely to cause higher healthcare costs despite its observational correlation. Thus, when using a genetic instrument, we would not identify a causal relationship between CRP and healthcare costs. In other words, a public health strategy that aims to lowering level of CRP would not lower healthcare costs. In epidemiology, Mendelian Randomization is typically used to identify the causal relationship between an exposure and a disease accounting for confounding and reverse causation [3]. We also used strong genetic instruments (e.g., F-statistic > 50) from well-powered genome-wide association studies (e.g., number of individuals included > 170,000).

Comment 3: The author used average health care cost of many years and risk factors at a given time. However, many risk factors such as WC, BMI, Blood pressure., glucose level, Hb A1c.....can change over time due to lifestyle change or clinical treatments. Changes in risk factors created a complex relationship between a risk factors and health care costs. I am not sure using an average health care cost of many years and an observed one-time risk factor in a regression model is appropriate.

Answer: We thank the reviewer for this comment. We agree that both exposures and healthcare costs can change over time, and may be affected by different types of interventions. An

important set of considerations is that our models already embody a degree of the longitudinal influence that you mention. First, Mendelian Randomization measures the cumulative effect of life-long exposure to a particular genotype, since these genotypes are established at conception and are borne by the individual throughout life. Second, healthcare costs are, as you note, a per-person measurement that reflects costs collected over several years, and therefore also do not represent "one-off" measurement costs, which for particular individuals may happen to be unusually high or low if only measured in a particular year. Third, the genetic variants were all identified in replicated genome-wide association studies. The genetic variants we use in the analyses have a known pattern of influence at a population level (e.g., a particular genetic variant may be associated with an increased risk of elevated blood pressure) and we rely on and interpret these associations in our analysis to make causal claims about the impact of a particular exposure on healthcare costs. Therefore, even given changes over time in the value of an exposure variable, we can expect that, on average over the population of interest, the effect of a genetic variant on an exposure will be as established in the underlying genome-wide association study.

This validity of these causal claims relies on the assumptions of Mendelian Randomization being respected. One of these assumptions is that we are measuring the local average treatment effect (LATE) of each exposure. The LATE is the average effect of an exposure on costs for individuals whose exposure was affected by the value of genetic instrumental variable(s) [5, 6]. Under this assumption, our estimates approximate the population-averaged causal effect rather than an individual exposure-outcome relationship, which may be non-linear. The consequence of the monotonicity assumption is that replacing a SNP not associated with (for example) blood pressure with a SNP associated with blood pressure would either increase blood pressure or leave blood pressure unchanged. This monotonicity assumption may be biologically plausible but is challenging to demonstrate empirically, because it requires comparison of the potential values of the exposure after replacing an individual's observed genotype with a hypothetical alternative genotype. Note that because for continuous exposures, it is reasonable to interpret the LATEs as reflecting the impact of the SNPs across the entire distribution of the exposure. We conduct our analysis and report its findings under the assumption that monotonicity holds for most if not all of our analysis sample. To acknowledge some of these points, the following has been added to the manuscript on page 19:

Second, our MR analysis was limited to linear instrumental variable estimates that approximate the population-averaged causal effect rather than an individual exposure-outcome relationship, which may be non-linear. The validity of the causal claims relies on the assumption that we are estimating the local average treatment effect (e.g., the average effect of a risk factor on healthcare costs for individuals whose exposure was affected by the value of the genetic instrumental variables). The linear instrumental variable estimates will be in the same direction as the causal effect for each individual in the population as long as the exposure-outcome relationship is monotonic. Thus, our MR analysis is valid in estimating the population-averaged causal effect but does not estimate the any non-linear exposure-outcome relationships for an individual.

Comment 4: Some of the results did not make intuitive sense. For example, diabetes is one of the most expensive diseases. Diabetes is defined as one who had a high a blood glucose level. It is hard to believe that high glucose or HbA1c would not lead to high health care cost.

Answer: We thank the reviewer for this comment. We agree that the lack of causal effect between HbA1c and healthcare costs - much like the null result between LDL cholesterol and healthcare costs that we highlighted in our discussion - prompts additional considerations. In the case of LDL cholesterol and healthcare costs, we hypothesized that the costs associated with preventative interventions (e.g., lipid-lowering medications) may counterbalance the costs associated with downstream morbidity (e.g., hospitalization). Similarly, in a population with well-managed diabetes, the impact of elevated HbA1c on healthcare costs may be lower than in a population that does not treat elevated HbA1c due to higher downstream costs. However, more advanced analyses that better explain cost reductions associated with certain medical interventions are certainly needed. To highlight this limitation, the following has been added to the manuscript on page 19:

Triangulation (Debbie A Lawlor, Kate Tilling, George Davey Smith, Triangulation in aetiological epidemiology, International Journal of Epidemiology, Volume 45, Issue 6, December 2016, Pages 1866–1886, <https://doi.org/10.1093/ije/dyw314>) with other evidence sources, such as randomized controlled trials, may better establish the mechanism through which medical services reduce specific categories of healthcare costs (e.g., whether preventative interventions indeed lower healthcare costs associated with downstream morbidity).

In addition, there are also at least two further conceptual issues regarding the interpretation of results. The first is that much of our knowledge on the association between risk factors/disease in relation to healthcare cost is drawn from conventional studies subject to the types of bias you mention above, such as omitted variables and simultaneity. We consider that it is important to avoid treating these existing observational associations as being more robust than they may be in truth, since this would suggest we have more knowledge of the causal effect of the variants on cost outcomes than we possess. The issue about accidents was precisely an example considered in our 2016 review paper where we considered what might be an “obesity related” cost [4]:

“This gives rise to a conceptual question: should the causal analysis of the cost consequences of obesity focus on total healthcare costs or on ‘obesity-related’ costs only? Casting the net widely to encompass total costs allows for unknown and unexpected influences on cost causally related to the variant and exposures of interest to be included in the analysis. Consider an example of an individual who experiences a car accident, to which diabetic retinopathy associated with obesity contributed, and who undergoes an expensive inpatient hospital stay. A focus on ‘obesity-related’ costs that excluded consideration of this type of emergency admission could overlook these costs, even though they are caused by obesity in the scenario described.”

The second conceptual issue is that our identification of causal effects relies on genetic variation. Therefore, our estimates are best understood as measuring the impact of “genetically influenced” exposures on healthcare costs. This may differ from other types of exposure on healthcare cost, and is an example of how Mendelian Randomization analyses does not necessarily comply with the “stable unit treatment value” assumption of causal inference, since it admits that different mechanisms of hypothetically manipulating an exposure (e.g., an individual may have well-controlled blood pressure because they take medication or they may have well-controlled blood pressure because they happen to have been born with a favorable genotype for blood pressure). To acknowledge this limitation, the following has been added to the manuscript on page 19:

Likewise, MR does not necessarily comply with the "stable unit treatment value" assumption of causal inference, as there may be different mechanisms of hypothetically manipulating an exposure. For example, an individual may have well-controlled SBP because they were born with a favorable genotype or because they are taking blood pressure medications, and the impact of these different influences on SBP may differ.

Reviewer #3 (Remarks to the Author):

The authors performed Mendelian Randomization (MR) to estimate the effects of 15 risk factors on annual total healthcare costs, using data from FinnGen (N=373,160), UK Biobank (N=307,048), and Netherlands Twin Register (N=16,726). Multiple MR methods were used to estimate the effects and stratified analyses (by service type, age, and sex) were performed. Multivariable MR analysis was also conducted to estimate the effect of risk factors on healthcare cost adjusted for major diseases. The major findings the authors showed were 1) significant effects of waist circumference, adult body mass index, and systolic blood pressure on increased annual total healthcare costs, 2) differences and consistency of these effects by service type, sex, and age, and 3) replication of these findings across Finland, UK, and Netherlands.

The methods in the study are generally sound and robust, and the MR findings are novel and of interest to not only the human genetics field but also sociogenomics and public health. I have only a couple of minor comments:

Comment 1. The description of running GWAS of healthcare costs in the UK Biobank seems to be missing? I could not find a description of how that GWAS was performed and the GWAS was not part of the original study by Dixon et al. Please highlight or add a description of the samples and methods used for the GWAS of healthcare costs in the UK Biobank.

Answer: We thank the reviewer for pointing this out. The GWAS was not described in the original study, so we have added the description. We have added the following to the manuscript on page 8:

For the UK Biobank, annual total healthcare costs were used as the primary outcome, in which pounds were converted to euros using the average exchange rate in 2021 of 1 EUR = 0.8403 GBP, and in-house GWAS protocols for quality control were used. Briefly, individuals with mismatch between genetic and report sex, individuals with sex chromosome aneuploidy, and related individuals of white British ancestry were excluded. The analysis was restricted to autosomal variants using a graded filtering process to account for imputation quality at different allele frequency ranges so that rarer genetic variants were required to have a higher imputation INFO score. Linear mixed models were implemented using BOLT-LMM and controlled for age, sex, and the first 40 principal components.

Comment 2. I understand the authors do not want to focus on the GWAS of healthcare costs but the MR analysis utilizing those GWAS. However, a separate section in the Methods for GWAS of healthcare costs in FinnGen (and for UK Biobank) is more appropriate than embedding it in the MR section.

Answer: We thank the reviewer for this suggestion. We have created a separate section in the Methods section for the GWAS of healthcare costs on page 6:

The primary outcome was log-transformed annual total healthcare costs, and our secondary outcomes included (1) log-transformed primary care, secondary care, and medication costs, (2) log-transformed annual total healthcare costs for females and males, and (3) log-transformed annual total healthcare costs for individuals under 30 years old, individuals between 30 and 60 years old, and individuals over 60 years old. We performed genome-wide association studies

(GWAS) of healthcare costs to identify genetic variants associated with healthcare costs using REGENIE, which is a method for fitting a whole-genome regression model.¹⁷ Briefly, REGENIE uses a two-step process that fits a whole-genome regression model and performs single-variant association testing. We used the default model with the following covariates: birth year, birth year squared, sex, 10 principal components, and batch covariates.

Comment 3. The quality of the GWAS of healthcare costs is not described anywhere in the paper. I could not find any description of basic GWAS summary statistics metrics such as lambdaGC or LD score regression intercept, significant association findings, Manhattan plot, Q-Q plot etc. It would be very helpful to provide the information for the reader to judge the quality of the GWAS (as a source data of the MR analyses).

Answer: We thank the reviewer for this suggestion to add more information about the GWAS results and we plan on making the summary statistics publicly available via the GWAS Catalog. Nonetheless, the focus of this paper is on the Mendelian Randomization analysis, and we have commenced related work, in collaboration with other studies, on a GWAS meta-analysis of healthcare costs for which full results and summary statistics will be made available. We have added the following text to the manuscript on page 10:

GWAS performed on log-transformed annual total healthcare costs identified several genome-wide significant SNPs (Supplementary Table 2 and Supplementary Figure 1 and 2).

We have also added a supplementary figure for the Manhattan plot, Q-Q plot, and a supplementary table for the top genome-wide significant hits (truncated for visibility).

Supplementary Figure 1. Manhattan plot from GWAS of log-transformed annual total healthcare costs in FinnGen.

Supplementary Figure 2. Q-Q plot from GWAS of log-transformed annual total healthcare costs in FinnGen. Lambda 0.7 = 1.304, lambda 0.5 = 1.293, lambda 0.1 = 1.285, lambda 0.01 = 1.29, lambda 0.001 = 1.295.

Supplementary Table 2. Top independent genome-wide significant hits from GWAS of healthcare costs in FinnGen.

chr	pos	rsid	ref	alt	pval	beta	se	eaf	info
1	24697088	rs144552623	A	G	7.77E-09	0.0641145	0.0111054	0.0126287	0.962229
1	113761186	rs6679677	C	A	6.92E-13	0.024785	0.00345146	0.146586	0.999818
1	182362628	rs11806150	A	G	2.19E-08	-0.0209685	0.00374699	0.121279	0.996973
2	27111756	rs2891550	C	T	4.14E-11	0.0165069	0.0025015	0.598099	0.990897
2	28211309	rs72818428	A	G	9.77E-10	0.0218303	0.00357105	0.135733	0.995658
...

Comment 4. The GWAS of healthcare costs are by themselves of interest to many fields as I mentioned above. I would suggest the authors to share the summary statistics publicly to facilitate the research fields (if this is not requested by the journal already).

Answer: We indeed plan on making the summary statistics publicly available via the GWAS Catalog.

Comment 5. I would encourage the authors to discuss the current findings of effects of the risk factors on healthcare costs with previous literatures of the same topics but based on conventional epidemiological approaches, or at least acknowledge previous studies on the same topic (but with different approaches) exist (or not?)

Answer: We thank the reviewer for this suggestion. We have added the following to the manuscript at page 19:

Previous studies have used conventional epidemiological approaches other than MR to quantify the effects of risk factors on healthcare costs. Hojgaard et al. used data from a Danish prospective cohort study and found that women with increased WC (≥ 80 cm) incurred \$261 more in annual healthcare costs than women with normal WC (< 80 cm) while men with increased WC incurred \$420 more in annual healthcare costs than men with normal WC. Pendergast et al. used data from an American and German prospective cohort study and found that individuals with higher WC have 16-18% (€300-€400) and 20-30% (\$1900-\$2400) higher healthcare costs compared to individuals with lower WC. Kirkland et al. used data from a nationally representative database from the United States and found that individuals with hypertension had \$1920 higher annual healthcare costs than individuals without hypertension.

Comment 6. There is a very minor conceptual point on the section title “Factors mediating the impact of risk factors on total healthcare costs”. The MVMR analysis described in this section does not provide an estimate of “the impact of risk factors on total healthcare costs” mediated through other major diseases (as suggested by the section title). MVMR provides estimates of risk factors’ effects on total healthcare costs adjusted/controlling for other major diseases. The text in this section actually used terms such as “accounting” and “adjusting”, not mediating. The authors may consider change the section title.

Answer: We have changed the section title to: "Impact of risk factors on total healthcare costs adjusted for risk of major diseases". We do indeed quantify the adjustment, rather than the mediation, and the main Mendelian Randomization analysis without adjustment remains the primary focus of our paper.

Comment 7. Information on blood pressure medications GWAS and instrument is missing?

Answer: We have added the information on blood pressure medication summary statistics to Supplementary Table 1.

Comment 8. For the MVMR analysis, the authors should consider (in the Discussion) that the imbalance of number of instruments (ranged from 1 to 52; ST1) for the five major diseases may affect the MVMR analysis results. The only attenuation effect observed were for BMI’s effect when adjusting for type 2 diabetes and for SBP when adjusting for blood pressure medications. This could just be a reflection of lack of power for the other diseases due to few instruments.

We thank the reviewer for highlighting this limitation. We have added the following to the manuscript at page 20:

Particularly, in our MVMR analysis, some mediators, such as stroke, had a small number of variants that likely affected statistical power. As more powerful GWAS are performed, MR may gain the statistical power to more accurately quantify causal effects.

References

- [1] Lagrand WK, Visser CA, Hermens WT, Niessen HW, Verheugt FW, Wolbink GJ, Hack CE. C-reactive protein as a cardiovascular risk factor: more than an epiphenomenon? *Circulation*. 1999 Jul 6;100(1):96-102. doi: 10.1161/01.cir.100.1.96. PMID: 10393687.
- [2] Sproston NR, Ashworth JJ. Role of C-Reactive Protein at Sites of Inflammation and Infection. *Front Immunol*. 2018 Apr 13;9:754. doi: 10.3389/fimmu.2018.00754. PMID: 29706967; PMCID: PMC590890
- [3] George Davey Smith, Shah Ebrahim, 'Mendelian randomization': can genetic epidemiology contribute to understanding environmental determinants of disease?, *International Journal of Epidemiology*, Volume 32, Issue 1, February 2003, Pages 1–22, <https://doi.org/10.1093/ije/dyg070>
- [4] Dixon, P., Davey Smith, G., von Hinke, S. et al. Estimating Marginal Healthcare Costs Using Genetic Variants as Instrumental Variables: Mendelian Randomization in Economic Evaluation. *PharmacoEconomics* 34, 1075–1086 (2016). <https://doi.org/10.1007/s40273-016-0432-x>
- [5] Burgess S, Davies NM, Thompson SG; EPIC-InterAct Consortium. Instrumental variable analysis with a nonlinear exposure-outcome relationship. *Epidemiology*. 2014 Nov;25(6):877-85. doi: 10.1097/EDE.000000000000161. PMID: 25166881; PMCID: PMC4222800.
- [6] Dixon P, Hollingworth W, Harrison S, Davies NM, Davey Smith G. Mendelian Randomization analysis of the causal effect of adiposity on hospital costs. *J Health Econ*. 2020 Mar;70:102300. doi: 10.1016/j.jhealeco.2020.102300. Epub 2020 Jan 25. PMID: 32014825; PMCID: PMC7188219.

Reviewer comments, second round

Reviewer #1 (Remarks to the Author):

The authors have addressed all my concerns and I don't have further comments.

Reviewer #3 (Remarks to the Author):

The authors have addressed all my comments. I have no further comments.

Reviewer #4 (Remarks to the Author):

This is a methodologically sound paper and from my point of view the concerns that reviewer 2 raised have been well addressed. The only concern/unclearity that still stands for me is point #3. In the study description it remains unclear when biomarkers were measured. Was this in average in the beginning, the middle or towards the end of the >20 year time period over which the outcome was measured. It also remains unclear in this context how the age stratification was done. Let's assume a person was 35 years at the time the biomarkers was measured and the costs were measured from age 20 to 40. In which age category was this person classified and with which arguments?

Next to this I identified a couple of other points that need some attention:

Line 244: "Briefly, individuals with mismatch between genetic and report sex, individuals with sex chromosome aneuploidy, and related individuals of white British ancestry were excluded." -> This sentence makes no sense to me.

Line 292: Mean "Total health care costs" in the study were quite low (2700€). Real total health care costs in Finland are probably rather around 4000€ Probably several cost categories (rehab, physio, aids and remedies etc.) are missing. If the term 'total health care costs' is used this should be discussed in the limitations.

Line 560: "Thus, our MR analysis is valid in estimating the population-averaged causal effect but does not estimate the any nonlinear exposure-outcome relationships for an individual." This inference from LATE to population-averaged causal effect is difficult to grasp. There is also a type in the sentence.

Line 237: Replication analyses in the United Kingdom and Netherlands. This section lacks detailed information to understand how this was exactly done. Aren't some of the exposure SNPs taken from UKBB GWAS? Maybe it would be helpful to add more information in a supplement section.

Line 168: Why were costs log-transformed? This makes interpretation of estimates more

difficult than necessary. With the sample size in FinnGen central limit theorem probably would justify the application of a linear model for the healthcare cost GWAS. See:

Mihaylova, B., Briggs, A., O'hagan, A., and Thompson, S. G. (2011). Review of statistical methods for analysing healthcare resources and costs. *Health Economics*, 20(8):897–916.

Line 429: Replication studies for the UK/UKBB: In a previous MR analysis using UKBB data, a 1 unit increase of BMI resulted in a 42 pound increase in total health care costs (Harrison et al. *Plos Med* 2021). In this study a 5 bmi unit increase resulted in a 103 pound increase in total health care costs. Despite the difference in methods (one sample vs. two sample etc.), it would be helpful to discuss this large difference (and the reasons for it) in estimates somewhere in the paper.

Line 446: The value of the PGS analyses in the study is not very well articulated to the reader.

Line 579: The paragraph before the conclusion statement contains quite general claims/points that are not very well linked to the presented study and results.

Comment 1: This is a methodologically sound paper and from my point of view the concerns that reviewer 2 raised have been well addressed. The only concern/unclear that still stands for me is point #3. In the study description it remains unclear when biomarkers were measured. Was this in average in the beginning, the middle or towards the end of the >20 year time period over which the outcome was measured. It also remains unclear in this context how the age stratification was done. Let's assume a person was 35 years at the time the biomarkers was measured and the costs were measured from age 20 to 40. In which age category was this person classified and with which arguments?

Answer: We thank the reviewer for the comment.

One core aspect of Mendelian Randomization is the use of genetic instruments to estimate the effect of exposure-outcome relationships, so we used the summary statistics from the GWAS of biomarkers and healthcare costs to estimate causal effects. In other words, Mendelian Randomization can be thought of as a “pseudo-randomized control trial” – take an example in which a subset of individuals in a cohort are assigned genotype A at birth. If it is found that there is a strong association between genotype A and biomarker X (estimated through GWAS), as well as between genotype A and healthcare costs (estimated through GWAS), there is genetically-informed evidence that biomarker X may have a causal effect on healthcare costs (estimated through Mendelian Randomization).

In this analysis we used GWAS results for several biomarkers to conduct Mendelian randomization. The biomarkers were measured at different ages depending of the original study. As many of the GWAS for biomarkers were conducted in UK Biobank, which included individuals between ages 40 and 69, most results refer to biomarkers measured during late adulthood.

Our assumption is the genetic effect on biomarker levels are stable across life. In other words, genetic variants that impact biomarkers levels in young ages are the same as those impacting biomarker at older ages. While environmental risk factors impacting biomarkers levels can differ across ages, we don't expect that to be the case for genetic, which is assigned at birth. Thus, the causal effects obtained from Mendelian randomization are the cumulative impact, over the lifecourse, of having a particular type of genotype.

Individuals were placed into age categories depending on the age at the end of follow-up. So, for example, if an individual was followed between age 20 to 40, their annual healthcare costs would be calculated by dividing the cumulative sum of healthcare costs between age 20 and 40 by their follow-up time, and they would be placed in the age category of “between 30 and 60 years old”. We have clarified the age definition in this sentence:

Methods (page 6): Annual total healthcare costs were the main outcome studied. In sensitivity analyses, we examined healthcare costs stratified by: (1) service type (e.g.,

primary care, secondary care, and medication costs), (2) sex, and (3) age at the end of follow-up (individuals under 30 years old, individuals between 30 and 60 years old, and individual over 60 years old).

Comment 2: Line 244: “Briefly, individuals with mismatch between genetic and report sex, individuals with sex chromosome aneuploidy, and related individuals of white British ancestry were excluded.” -> This sentence makes no sense to me.

Answer: We thank the reviewer for this comment. These types of exclusions are typically done in genetic studies for the following reasons. First, mismatch between genetic sex and self-reported gender is likely to reflect sample mismatch. This is a standard quality control step advised by UK biobank. Second, limitation to Europeans ancestry is done because the original GWAS used to derive the instrumental variable for mendelian randomization were mostly conducted in European ancestries. We have clarified this sentence:

Methods (page 8): *Briefly, the following exclusion criteria were applied: (1) individuals with mismatches between their biological sex and self-reported gender, (2) individuals with sex chromosome aneuploidy, and (3) related individuals of white European British ancestry.*

Comment 3: Line 292: Mean “Total health care costs” in the study were quite low (2700€). Real total health care costs in Finland are probably rather around 4000€. Probably several cost categories (rehab, physio, aids and remedies etc.) are missing. If the term ‘total health care costs’ is used this should be discussed in the limitations.

Answer: We thank the reviewer for this comment. There are certainly cost categories that are not captured by our cost estimates. Specifically, given the total Finnish budget of €21,054M, we do not capture private occupational healthcare (€748M), institutional care for elderly and disabled individuals outside of a hospital setting (€3882M), privately provided healthcare and rehabilitation reimbursed by Kela (€1089M), eyeglasses and other medical devices (€488M), healthcare-related transportation costs such as ambulances and reimbursed taxis (€356M). Once these cost categories are removed from the total budget, the per capita expenditures are approximately €2,390 per year per person, which is consistent with the cost estimates that we derive from the registries in our study. We have added the following to reflect this:

Methods (page 5): *We did not capture costs related to private occupational healthcare, institutional care for elderly and disabled individuals outside of a hospital setting, eyeglasses and other medical devices, and healthcare-related transportation costs such as ambulances and reimbursed taxis.*

Discussion (page 21): *Fifth, while we comprehensively capture most healthcare costs in our cost estimates through using registry-based data, there are certain cost categories that are not captured by national healthcare registries (e.g., private occupational healthcare, institutional care for elderly and disabled individuals outside of a hospital setting, eyeglasses and other medical devices, and healthcare-related transportation costs such as ambulances and reimbursed taxis). As such, our cost estimates are representative of primary outpatient, secondary and tertiary inpatient and outpatient hospital visits, and medication costs.*

Comment 4: Line 560: “Thus, our MR analysis is valid in estimating the population-averaged causal effect but does not estimate the any nonlinear exposure-outcome relationships for an individual.” This inference from LATE to population-averaged causal effect is difficult to grasp. There is also a type in the sentence.

Answer: We thank the reviewer for this comment. We agree that assuming monotonicity implies that the local average treatment effect is being estimated (e.g., for individuals whose value of the exposure were either increased or unchanged by having the allele associated with increases in the value of the exposure) and the average causal effect may include individuals that deviate from this definition. As such Mendelian Randomization estimates may be closer to the local average treatment effect. We have clarified this sentence:

Discussion (page 20): Thus, our MR analysis is valid in estimating the local average treatment effect of a risk factor on healthcare costs in our cohort rather than any non-linear effects between exposure and outcome for any single individual.

Comment 5: Line 237: Replication analyses in the United Kingdom and Netherlands. This section lacks detailed information to understand how this was exactly done. Aren't some of the exposure SNPs taken from UKBB GWAS? Maybe it would be helpful to add more information in a supplement section.

Answer: We thank the reviewer for this comment. We have updated the following to better describe the replication analyses in the United Kingdom and Netherlands (edits underlined):

Methods (page 9): *We repeated our analyses to estimate the annual monetary impact per capita associated with each risk factor. We applied two-sample MR using largely the same summary statistics for risk factors and summary statistics from the UK Biobank¹⁰ and Netherlands Twin Register²⁷ for healthcare costs. For the risk factor summary statistics that included individuals from the UK Biobank, we used separate summary statistics to ensure non-overlapping summary statistics between exposures and outcomes (Supplementary Table 1). We also calculated the genetic correlation between the three sets of summary statistics (e.g., GWAS on healthcare costs in Finland, United*

Kingdom, and Netherlands) to evaluate the consistency of associations across different healthcare systems using linkage disequilibrium score regression (LDSC), a tool to estimate genetic correlation and heritability.³¹ Briefly, after filtering (e.g., INFO > 0.9 and minor allele frequency (MAF) > 0.01), LDSC with default parameters and Hapmap3 SNPs in the 1000 Genomes Project European reference panel was used.³¹

We also constructed polygenic scores (PGS) of healthcare costs from the UK Biobank and Netherlands Twin Register and estimated their associations with healthcare costs in FinnGen. Using the GWAS summary statistics from the UK Biobank and Netherlands Twin Register to construct PGS in the FinnGen cohort allowed us to compare whether the healthcare costs estimated using genetic associations in independent cohorts were consistent with the registry-based estimates of healthcare costs. In other words, a strong correlation between the PGS-predicted costs in two independent cohorts and registry-based costs in Finland would indicate consistency and validity of the genetic associations for healthcare costs across multiple countries. We first used PRS-CS 32 to calculate weights of association and PLINK2³³ to calculate scores. Briefly, PRS-CS is a polygenic prediction method that uses Bayesian regression and infers posterior SNP effect sizes under continuous shrinkage priors using only GWAS summary statistics and an external linkage disequilibrium reference panel.³² The 1000 Genomes reference panel was used to output weights using standard PRS-CS parameters. Only HapMap3 variants were included. PLINK2 was used to calculate the PGS, which were then standardized across the entire FinnGen Study cohort with a mean of 0 and a standard deviation of 1.

Comment 6: Line 168: Why were costs log-transformed? This makes interpretation of estimates more difficult than necessary. With the sample size in FinnGen central limit theorem probably would justify the application of a linear model for the healthcare cost GWAS. See: Mihaylova, B., Briggs, A., O'hagan, A., and Thompson, S. G. (2011). Review of statistical methods for analysing healthcare resources and costs. Health Economics, 20(8):897–916.

Answer: We thank the reviewer for this comment. In addition to the fact that healthcare costs are frequently log-transformed to address its right-skewness, we chose the log transformation in order to interpret the estimates as the percent change in healthcare costs per one standard deviation increase in the exposure. In other words, we use a log-linear model such that estimates can be back-transformed and interpreted as percents rather than raw euro values. For example, after log-transformation, if $\beta = 0.1$, this β can be back-transformed as $100 \cdot (e^{0.1} - 1) = 10.5\%$, which is interpreted as one standard deviation increase in the exposure yielding a 10.5% change in the outcome. We found that this interpretation as a percent is more interpretable than raw euro values as different countries utilize different currency systems, have different magnitudes of healthcare expenditure, etc. We have added the following to reflect this:

Methods (page 6): *Log transformation is frequently applied to normalize right-skewed healthcare costs, although certain limitations remain (e.g., handling zero cost estimates and requiring an adjustment such as $\log(X + 1)$).¹⁷⁻²⁰ Log transformation was also used to make effect estimates more interpretable as effect estimates calculated from log-transformed outcomes yields percent changes (e.g., one standard deviation increase in the dependent variable yields a certain percent change in the outcome). Percent changes, rather than raw euro values, may be more interpretable as different countries utilize different currency systems, have different magnitudes of healthcare expenditure, etc.*

Comment 7: Line 429: Replication studies for the UK/UKBB: In a previous MR analysis using UKBB data, a 1 unit increase of BMI resulted in a 42 pound increase in total health care costs (Harrison et al. Plos Med 2021). In this study a 5 bmi unit increase resulted in a 103 pound increase in total health care costs. Despite the difference in methods (one sample vs. two sample etc.), it would be helpful to discuss this large difference (and the reasons for it) in estimates somewhere in the paper.

Answer: We thank the reviewer for this comment. Harrison et al. 2021 used costs from the UK Biobank capturing both primary and secondary care costs, in which the primary costs needed to be calculated using multiple imputation given incomplete coverage of these costs in UK Biobank. To permit comparability with the complete Finnish data, for which we do not require the use of multiple imputation, we used the summary statistics for costs in the UK Biobank from Dixon et al. 2020, which only used inpatient hospital costs. As such, the Harrison et al. estimates are larger than the estimates from both our study and Dixon et al. Our estimate of a £103 healthcare cost increase for 5 BMI units is consistent with the estimate from Dixon et al. of a £21 healthcare cost increase for 1 BMI unit. Throughout our manuscript, we cite Dixon et al. as the source of our summary statistics.

Comment 8: Line 446: The value of the PGS analyses in the study is not very well articulated to the reader.

Answer: We thank the reviewer for this comment. We have added the following to clarify the importance of the PGS analysis:

Methods (page 9): *Using the GWAS summary statistics from the UK Biobank and Netherlands Twin Register to construct PGS in the FinnGen cohort provided an additional approach on top of genetic correlation to understand if genetic signals in one study could replicate in the other studies, despite difference in the healthcare costs definitions across countries.*

Discussion (page 16): *Overall, despite lack of genome-wide significant signals in UK biobank and the Netherlands Twin Register, we could derive a PGS for healthcare costs*

with a significant effect in FinnGen suggesting consistency in genetic associations for healthcare costs across multiple countries.

Comment 9: Line 579: The paragraph before the conclusion statement contains quite general claims/points that are not very well linked to the presented study and results.

Answer: We thank the reviewer for this comment. We have modified this paragraph to better reflect the potential applications of our approach and findings:

Discussion (page 21): *Our approach might inform the cost-effectiveness of common healthcare screening procedures based on biomarkers measurement. More in general, linking genetics to healthcare costs opens different research venues. For example, evaluation of the costs associated with specific genetic variants that mimic drug targets may inform drug development and commercialization. Implementation of genetic screening either in the form of polygenic score or single variants, would require health-economic assessment.⁴⁸⁻⁵⁰ Future large-scale genetic studies will be powered to provide a comprehensive assessment on the impact of genetics on healthcare costs and facilitate the implementation of such proposed genomic medicine approaches.*

Reviewer comments, third round

Reviewer #4 (Remarks to the Author):

Thank you very much.

All my concerns/points have been solved and adressed in a sufficient manner.

REVIEWERS' COMMENTS

Reviewer #4 (Remarks to the Author):

Thank you very much.

All my concerns/points have been solved and adressed in a sufficient manner.

We thank the reviewer for their feedback.